# Predicting the water film depth: A model based on the geometric features of road and capacity of drainage facilities

**Shuo Han**[1], **Jinliang Xu**[2]*, **Menghua Yan**[2], **Sunjian Gao**[2], **Xufeng Li**[2], **Xunjiang Huang**[2], **Zhaoxin Liu**[3]

**1** College of Transportation Engineering, Chang'an University, Xi'an, China, **2** School of Highway, Chang'an University, Xi'an, China, **3** Shandong Hi-speed Infrastructure Construction Co., Ltd., Jinan, China

* xujinliang@chd.edu.cn

**Data Availability Statement:** All relevant data are within the manuscript and its Supporting Information files.

## Abstract

The water film depth is a key variable that affects traffic safety under rainfall conditions. According to the Federal Highway Administration, approximately 5700 people are killed and more than 544 700 people are injured in crashes on wet pavements annually. While several studies have attempted to address water film depth issues by establishing prediction models, a few focused on the relationship among road geometric features, capacity of drainage facilities and water film depth. To ascertain the influence of the geometric features of road and facility drainage capacities on the water film depth, the road geometry features were first classified into four types, and the facility drainage capacities were considered from three aspects in this study. Furthermore, the concept of short-time rainfall grade was proposed according to the results of the field test. Finally, the theoretical prediction model for the water film depth was conceived, based on the geometric features of road and facility drainage capacities with different rainfall intensities. Compared with the traditional regression prediction models, the theoretical prediction model clearly shows the effects of the geometric features of road and facility drainage capacities. When the road drainage facilities have no drainage capacity, the water film depth increases rapidly with the rainfall intensity. This model can be used to predict the water film depth of road surfaces on rainy days, evaluate the effect of rainfall on the driving environment, and provide guidance for determining safety control measures on rainy days.

## 1. Introduction

Studies have demonstrated that the wet and slippery nature of roads, accompanied with the decrease in friction, caused by rainfall is a key variable in traffic accidents [1–5]. Evidently, with less friction, the braking distance of the vehicle increases at the same speed, which triggers frequent collisions in rainy weather. In their research, Brodsky et al. demonstrated that the accident rate on wet and slippery roads on rainy days was 2–3 times higher than that on dry roads on sunny days. Moreover, the longer stopping sight distance required due to a reduction in friction between tiers and the wet road surface might be more challenging on curves [3]. To

**Funding:** The author(s) received no specific funding for this work.

**Competing interests:** The authors have declared that no competing interests exist.

further study the reasons for the reoccurring rain collision accidents caused by the reduction in the friction coefficient, researchers focused on the relationship between the friction coefficient and the water film depth, and reached the consensus that there was a direct correlation between the reduction in friction coefficient and the water film depth on road surface [6–8]. Studies inferred that there are primarily two methods of regression analysis and simulation adopted in studying the relationship between the friction coefficient and the water film depth. Using artificial rainfall, Luo et al. carried out dynamic friction coefficient test experiments under different water film depth and determined that the reduction in the friction coefficient is directly related to the water film depth of the road surface [6]. In addition, via indoor water spray experiments, Do et al. determined that the relationship between the friction coefficient and the water film depth (less than 1 mm) agrees with the Stribeck curve [7]. Unlike regression analysis, the finite element simulation model was established to predict the friction coefficient under different water film depths, and it was deduced that at the same driving speed, the friction coefficient decreased with the increase in the water film depth [8]. Although the research methods differ, they all arrived at a consistent conclusion: The thicker the water film, the lower the friction coefficient. Further, the hydroplaning phenomenon may occur, which adversely impacts driving safety under rainfall conditions [9–11].

Hence, a prediction model for the water film depth is crucial in determining the road friction coefficient and evaluating the impact of rainfall on traffic safety. However, various geometric features of road and facility drainage capacities, as well as different rainfall intensities, exert different influences on the water film depth. Luo et al. adopted the variance analysis and multi-factor treatment statistical methods and concluded that the calculation of water film depth is affected by rainfall (rainfall intensity and duration) and pavement characteristics (cross slope, longitudinal slope, pavement material properties, texture, and penetration) [12]. Regarding the aforementioned complex factors, several investigations on the water film depth have been conducted, such as regression analysis and artificial neural network [13–16]. Among them, the most widely adopted method is the water film depth regression model established via experiments. For example, based on the in-door rainfall simulation experiment, the UK Road Research Laboratory proposed the conventional RRL model, i.e., the rain water depth relative to drainage length, rainfall intensity (5-min duration), and slope [13]. Anderson modified the parameters of the RRL model and proposed a water film depth prediction model for a plane impermeable surface [14]. Unlike the RRL model, which only considers two pavement texture depths (1.8 mm and 2.4 mm), Gallaway developed a water film depth prediction model for the U.S. Department of Transportation, which considers nine pavement texture depths. The relationship between rainfall intensity, cross slope, surface texture, drainage-path length, and water film depth are determined through artificial rainfall experiments [15]. The RRL, Anderson, and Gallaway models are simple and widely used by other researchers for developing water film depth prediction models [17, 18], regression model verification [19, 20], water film depth prediction [21–23], etc.

Clearly, the aforementioned regression models are in accordance with some specific test data in their experience. Their applications are limited to the types of parameters covered by the models, and when the parameter values are within the scope of the database used in their derivation. Owing to the changeable combination of road slope, superelevation, and alignment, facility drainage capacities and environment, the regression model may not accurately describe the influence of various factors on the pavement water film depth.

To address the limitations of the above approaches, this study intends to establish a prediction model for the water film depth, with a full theoretical foundation and extensive application scope to clearly demonstrate the effect of the main parameters. The theoretical model for predicting the water film depth under different rainfall intensities is established by considering

four geometry features of roads and three kinds of facility drainage capacities to predict the relationship among the water film depth, the geometric characteristics of the road and the facility drainage capacities accurately.

Compared with the regression models, the theoretical model established in this study considers the geometric characteristics of the real road environment and the drainage capacity of drainage facilities; therefore, it can accurately reveal the main factors affecting the water film depth and describe the influence of the rainfall intensity, road geometric characteristics, and facility drainage capacities on the water film thickness. Hence, the proposed model on a more theotical basis overcomes the limitations of regression models.

The rest of this paper is organized as follows. Section 2 presents the construction of the mathematical model for the water film depth, including the calculation of rainfall, development of water accumulation model, calculation of displacement of drainage facilities, and derivation of formulas for road surface water film depth. Section 3 presents an analysis of the results obtained, where the theoretical model for water film depth is verified, and the effects of different parameters on water film depth are discussed. Section 4 presents the conclusions, discussions, and prospects for further research.

## 2. Mathematical model

When water accumulates on road surface, a water film is formed with a certain depth, which provides the basic concept of the water film depth prediction model. The amount of accumulated water is defined as the amount of rainfall minus the amount of drainage, which means that the volume of surface-gathered water in unit time is the difference between the volume of rainfall and drainage. Even if the same volume of water is gathered, the different road geometry features and convergence directions of flow cause different water film depths. The amount of rainfall and drainage are related to the rainfall intensity and capacity of drainage facilities. Therefore, in this study, we propose a prediction model for water film depth based on the geometric features of roads and capacity of drainage facilities with different rainfall intensities.

For field verification and model derivation, the pavement water film depth in rainy days mentioned here is defined as the depth of water film under a short-time (1 h) rainfall. Compared with the unorganized water without water retaining curb stones such as the Hangshaotai Expressway, rainfall converges on the road surface, forming a overland flow and then discharged from the drainage facilities with a water-retaining belt or curb stone. Therefore, this study focuses on the water film depth at the water retaining belts or curb stones. The road surface water film depth is related to the amount of accumulated water, which refers to the difference between the amount of rainfall and displacement of drainage facilities. Various road alignments and combinations result in different water film depths, although with the same water accumulation. Hence, the construction of water film depth model includes the calculations on rainfall, construction of the water accumulation model, calculation of the displacement of drainage facilities, and derivation of formulas on the water film depth of road surfaces.

To improve the applicability of the model, this study establishes a theoretical water film-thickness prediction model based on different geometric characteristics and drainage capacities of actual roads, ignoring the effects of rainfall evaporation, infiltration and runoff, and pavement material properties.

### 2.1. Rainfall

This study focuses on rainfall scenarios with water retaining belts or curb stones. Owing to the different combinations of road alignment, water retaining belts or curb stones, rainfall will be

collected on the highway surface, which formed a overland flow, was accumulated and then discharged from the drainage facilities. The amount of rainfall is equal to the volume of converged slope flow per unit time.

In this research, the amount of rainfall is defined as the volume of rainwater accumulated on the pavement surface without evaporation, infiltration, and runoff, which comes from the rainwater falling from the sky to the ground in the case of a certain catchment area; the unit is m³/h. Rainfall is the product of the rainfall intensity and catchment area. The catchment area is related to the composite slope of the road and the configuration of drainage facilities. The relationship model between rainfall and rainfall intensity is expressed as follows:

$$Q = I_R F \times 10^{-3} = I_R L W \times 10^{-3} \tag{1}$$

where the $Q$, $I_R$, $F$, $L$, and $W$ represent the amount of rainfall (m³/h), rainfall intensity (mm/h), catchment area (m²), catchment length (m), and catchment width (m), respectively.

## 2.2. Road surface water accumulation

The reason for water gathering on a surface is attributed to road sections with poor drainage. The longitudinal gradient should not be less than 0.3%. Regarding the superelevation transition, the resultant gradient should not be selected as 0%. In addition, to ensure the smooth drainage of the pavement, comprehensive drainage facilities should be selected when the resultant gradient is less than 0.5% [24]. In this research, the road section with cross slop less than 0.3% is regarded as the section with poor drainage.

Even though the same volume of water is gathered, the different road geometry features and convergence directions of flow ensure different water film depths. According to the gradient values of cross and longitudinal slopes, the sections with poor drainage are divided into four types. Correspondingly, from the geometric perspective, four geometric models for road surface water are established by the AutoCAD software, as presented in Table 1. Road surface water accumulation and water film depth refer to the volume and height of water, respectively, in the geometric model.

The geometric models of water film depths with different road geometric characteristics are established based on different combinations of cross and longitudinal slopes. To obtain the volume and height of the geometric model, i.e., road surface water accumulation and water film depth, it is necessary to further derive the volume and water depth calculation formulas of different geometric models. Subsequently, the derivation process of the mathematical formula is described in detail.

**2.2.1. Models A & B.** When the cross slop is $\geq 0.3\%$, and the longitudinal gradient is located in the range of -0.3% to +0.3%, this indicates a road section with poor longitudinal drainage, and the drainage is mainly carried out by the cross slop. Both the circular curve and

**Table 1. Geometric models of road water accumulation.**

| Model Number | Type | Index | Schematic Illustration |
|---|---|---|---|
| A | Road section with optimal cross drainage and poor longitudinal drainage (Poor drainage section at longitudinal gradient with the slope change point of a concave vertical curve) | cross slop ≥ 0.3% and longitudinal gradient -0.3%–+0.3% | Fig 1 |
| B | Road section with optimal cross drainage and poor longitudinal drainage (Poor drainage section at longitudinal gradient without the slope change point of a vertical curve) | cross slop ≥ 0.3% and longitudinal gradient -0.3%–+0.3% | Fig 4 |
| C | Road section with poor cross and longitudinal drainages | cross slop -0.3%–+0.3% and longitudinal gradient -0.3%–+0.3% | Fig 5 |
| D | Road section with poor cross drainage and optimal longitudinal drainage | cross slop -0.3%–+0.3% and longitudinal gradient ≥0.3% | Fig 6 |

parabola can be adopted to describe the vertical curve. In general, there is a negligible difference between a circular curve and a parabola for the highway design. Here, the circular curve functions as the alignment of the vertical curve to easily establish the geometric model. The road surface water accumulation models can be divided into two categories based on the absence or presence of the slope change point of the concave vertical curve involved in a road section with poor longitudinal drainage.

*2.2.1.1. Model A.* When the slope change point of the concave vertical curve is involved in a road section with poor longitudinal drainage, the road surface water accumulation model fits a scalene cylinder, where the volume of the accumulated water is equivalent to the volume of water in the scalene cylinder, and the depth of road surface water film is equal to the water depth at the bottom of the scalene cylinder (*H*). Fig 1 schematically illustrates the water accumulation Model A for the road section with optimal cross drainage and poor longitudinal drainage.

For ease of calculation, assuming that a cylinder with a bottom radius *R* is cut by a plane at an angle *β*, the intercepted cylinder is an inclined cylinder. Hence, a bow height *H* is formed at the bottom, as illustrated in Fig 2. In addition, the bottom of the cylinder is presented in Fig 3. It can be inferred that the depth of the road surface water film is the bow height *H*, and the volume of the accumulated water is equivalent to the volume of the scalene cylinder cut by the cross section.

When the section is cut by a cross section plane perpendicular to the *x*-axis, each section shared with a right triangle can be obtained, and the function of the cross-sectional area $A(x)$ can be defined as

$$A(x) = \frac{1}{2} BD \cdot BF = \frac{1}{2} BD^2 \tan\beta \tag{2}$$

$$BD = BG - DG = \sqrt{R^2 - x^2} - OC = \sqrt{R^2 - x^2} - (R - H) \tag{3}$$

$$\text{Therefore, } A(x) = \frac{1}{2} \left[ \sqrt{R^2 - x^2} - (R - H) \right]^2 \tan\beta \tag{4}$$

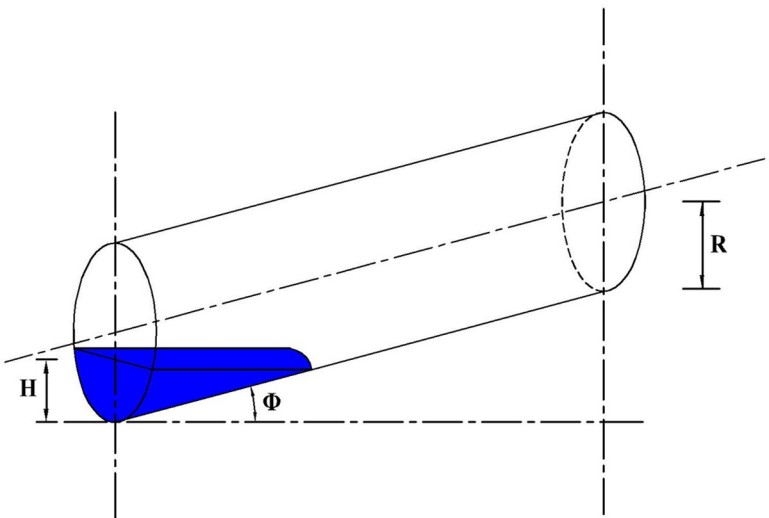

**Fig 1. Road surface water accumulation Model A.**

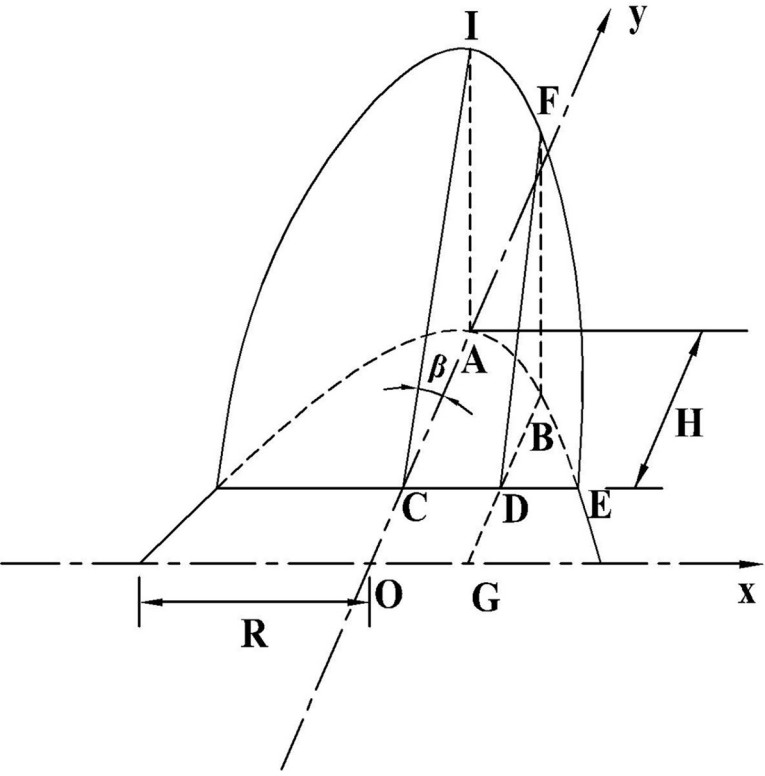

**Fig 2. Geometric diagram of a scalene cylinder cut by cross section.**

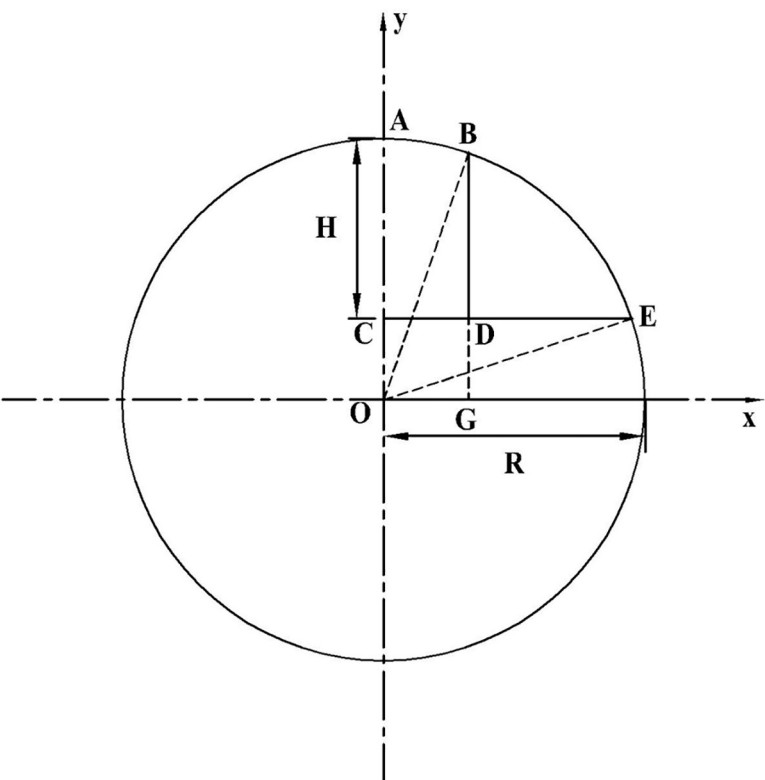

**Fig 3. Geometric diagram of the bottom of the cylinder.**

where $A(x)$, $R$, $H$, $x$, and $\beta$ represent the cross-sectional area (m$^2$), radius of the vertical curve (m), water film depth (m), vertical distance between cross section BDF and the y-axis (m), and angle of cross section CEI to the bottom (°), respectively.

The volume of the accumulated water on a road surface, i.e., the volume of the inclined cylinder cut by the cross section $V_{j1}(H)$, can be derived as follows:

$$
\begin{aligned}
V_{j1}(H) &= 2\int_0^{\sqrt{R^2-(R-H)^2}} \frac{1}{2}\left[\sqrt{R^2-x^2}-(R-H)\right]^2 \tan\beta \, dx \\
&= \left\{\sqrt{R^2-(R-H)^2}\left[\frac{2}{3}R^2+\frac{1}{3}(R-H)^2\right]-R^2(R-H)\arcsin\frac{\sqrt{R^2-(R-H)^2}}{R}\right\}\tan\beta \\
&= \left\{\sqrt{R^2-(R-H)^2}\left[\frac{2}{3}R^2+\frac{1}{3}(R-H)^2\right]-R^2(R-H)\arcsin\frac{\sqrt{R^2-(R-H)^2}}{R}\right\}\frac{1}{i_h}
\end{aligned}
\tag{5}
$$

where $V_{j1}(H)$, $R$, $H$, and $i_h$ represent the volume of the accumulated water on road surface (m$^3$) with Model A, radius of the vertical curve (m), depth of water film (m), and cross slope, respectively.

*2.2.1.2. Model B*. When the slope change point of the vertical curve is not involved in a road section with poor longitudinal drainage, the section of the poor longitudinal drainage is located on the side of the vertical curve or straight slope section. Owing to the optimal cross drainage and poor longitudinal drainage, road surface water will accumulate along the longitudinal slope of the pavement. In this case, the model of the road surface water accumulation fits a triangular prism, as illustrated in Fig 4. Here, the y- and x-axes denote the outer boundary line and the real-time length of the road accumulated water, respectively, whereas $H$ is the real-time depth of the water film. In this case, the volume of the accumulated water on the road can be calculated using the following formula:

$$
V_{j2}(H) = \frac{1}{2}xHW = \frac{1}{2}\frac{H^2}{\tan\beta}W = \frac{H^2W}{2i}
\tag{6}
$$

where $V_{j2}(H)$, $L$, $W$, $H$, and represent the volume of the accumulated water on the road surface (m$^3$) with Model B, catchment length (m), catchment width (m), depth of water film outside the road, and longitudinal slope, respectively.

**2.2.2. Model C.** When both the cross and longitudinal gradients are located in the range of -0.3% to +0.3%, the cross slope and longitudinal slope of the road exhibit poor drainage. On the road section with superelevation transition, there should be a section of zero slope section outside the carriage way. If the gradient rate of superelevation outside the carriage way is small, i.e., significantly less than 1/330, the road section with the small cross slop will become longer, which leads to a longer road section with poor cross drainage. Simultaneously, if the section also has a negligible longitudinal gradient ($\leq 0.5\%$), a small resultant gradient will be obtained correspondingly, thereby generating poor drainage in the entire road section. At this point, the road water accumulation model fits a horizontal cylinder, as illustrated in Fig 5, and it is defined as Model C. The shadow area $S$ presented in Fig 5 is equal to the area difference between sector and triangle, which is expressed as

$$
S = 2\left[\frac{1}{2}\alpha R^2 - \frac{1}{2}(R-H)R\sin\alpha\right]
\tag{7}
$$

where $S$, $\alpha$, $R$, and $H$ represent the shadow area (m$^2$), central angle of the sector (rad), radius of

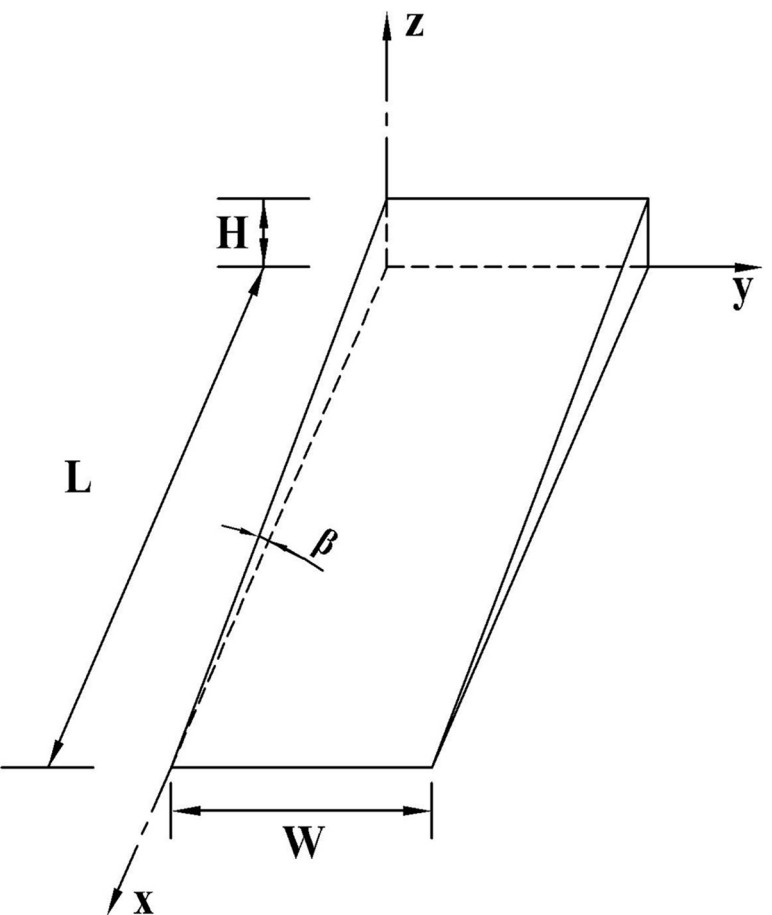

**Fig 4. Road surface water accumulation Model B.**

the vertical curve (m), and depth of water film (m), respectively.

$$\alpha = \arccos\frac{R - H}{R} \tag{8}$$

The volume of the accumulated water on road $V_{j3}(H)$ can be described as

$$V_{j3}(H) = SW = \left[R^2\arccos\frac{R - H}{R} - (R - H)\sqrt{R^2 - (R - H)^2}\right]W \tag{9}$$

where $V_{j3}(H)$, $R$, $W$, and $H$ represent the volume of the accumulated water on the road surface (m³) with Model C, radius of the vertical curve (m), catchment width (m), and water film depth, respectively.

**2.2.3. Model D.** When the cross slops are located in the range of -0.3% to +0.3%, and the longitudinal gradient is more than 0.3%, a small resultant gradient will be formed. Accordingly, the retention time of rainwater on the road surface will be longer; hence, drainage will be achieved through the longitudinal slope of the road. In this case, the road water accumulation model fits the triangular prism shape. As illustrated in Fig 6, the $y$- and $x$-axes represent the outer boundary line and real-time width of the accumulated water, respectively, and $H$ is the real-time depth of the water film. Therefore, the volume of the accumulated water on road

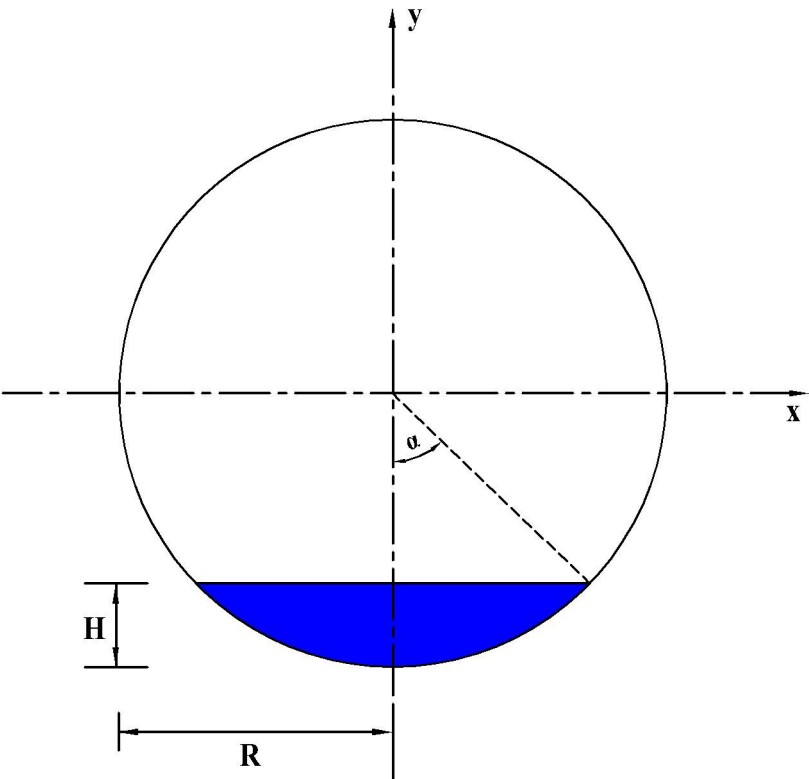

**Fig 5. Road surface water accumulation Model C.**

$V_{j4}(H)$ can be calculated using the following formula:

$$V_{j4}(H) = \frac{1}{2}xHL = \frac{1}{2}\frac{H^2}{\tan\alpha}L = \frac{H^2L}{2i_h} \tag{10}$$

where the $V_{j4}(H)$, $L$, $W$, and $H$ represent the volume of the accumulated water on road surface (m³) with Model D, catchment length (m), catchment width (m), and depth of water film, respectively.

## 2.3. Water displacement of drainage facilities

The amount of pavement drainage is related to the drainage capacity per unit time. During rainfall, the depth of water film on road can be directly affected by the drainage capacity of the road surface. Water from the rainfall is collected on the surface of the road, which facilitates the formation of the slope flow. After setting the water retaining belt or curb stone, the flow section of the slope flow takes a shallow triangle shape with a single cross slope. The calculation of the displacement adopts the shallow triangular discharge capacity formula of a single cross slope specified by the Specifications for Drainage Design of Highway (JTG/T D33-2012) [25], and it is expressed as

$$Q_c = 0.377\frac{1}{i_h n}h^{\frac{8}{3}}I^{\frac{1}{2}} \tag{11}$$

where $Q_c$, $i_h$, $h$, $n$, and $I$ represent the discharge capacity of a shallow triangle with the single cross slope (m³/s), cross slope of flow section, depth of water of flow section (m), roughness

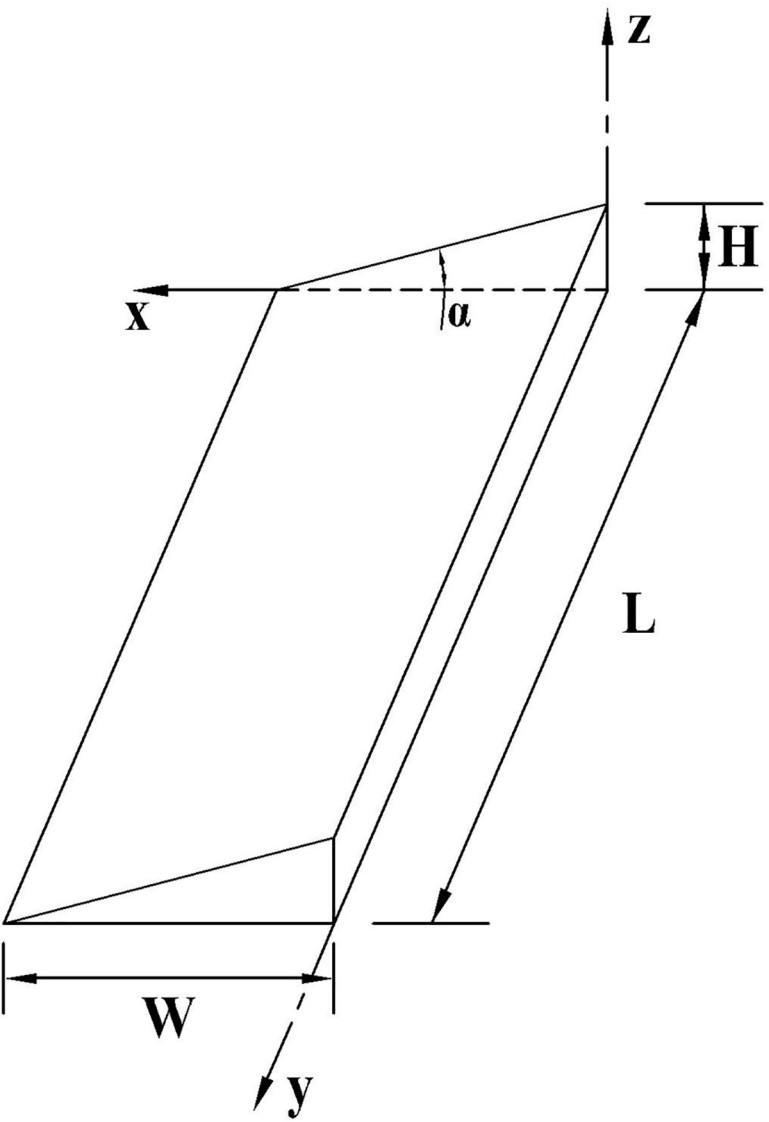

**Fig 6. Road surface water accumulation Model D.**

coefficient of trench wall or pipe wall (0.013 for asphalt pavement), and hydraulic gradient, i.e., longitudinal gradient of road, respectively.

## 2.4. Calculation of water film depth

The water film depth is related to rainfall, water accumulation, and amount of drainage. In addition, the drainage amount depends on the capacity of road drainage facilities. Considering measures such as the presence or absence of road drainage facilities, as well as the capacities of these drainage facilities, the water film depth can be calculated in three situations, which includes the road with drainage facilities and in normal use, the road with drainage facilities of poor capacities, and the road without drainage capacities. The derivation process of the water film depth calculation formula corresponding to these three situations will be comprehensively described.

**2.4.1. Road with drainage facilities and in normal use.** When the road is equipped with drainage facilities, and these facilities are in their normal use, the water displacement from drainage facilities $Q_c$ can be greater than or equal to the amount of rainfall $Q$. The rainfall can be completely discharged and there will be no water accumulation on the road surface; hence, $Q$ is equal to $Q_c$. During rainfall, the depth of water at the flow section is the real-time depth of the pavement's water film. Owing to the different units of rainfall and drainage, the relationship between drainage and rainfall after unit conversion is obtained as

$$3600Q_c = Q \tag{12}$$

where $Q_c$ and $Q$ refer to the amount of water displacement from drainage facilities (m$^3$/s) and the amount of rainfall (m$^3$/h), respectively.

The real-time water film depth of the road surface during rainfall is the water depth of the cross-section. From Eq 12, the water film depth model can be derived as

$$H = 0.005\left(\frac{I_R L W i_h n}{I^{\frac{1}{2}}}\right)^{\frac{3}{8}} \tag{13}$$

where $H$, $I_R$, $L$, $W$, $i_h$, $n$, and $I$ represent the water film depth (m), rainfall intensity (mm/h), catchment length (m), catchment width (m), cross slope of flow section, roughness coefficient of trench wall or pipe wall (0.013 for asphalt pavement), and hydraulic gradient, i.e., longitudinal gradient of road, respectively.

**2.4.2. Road with drainage facilities of poor capacities.** When the road has drainage facilities with poor drainage capacities, the amount of rainfall $Q$ will be greater than water displacement from these drainage facilities $Q_c$. Therefore, the rainfall cannot be completely discharged, thereby triggering water accumulation on the road. The designed water displacement of drainage facility is marked as $Q_{cmax}$. The theoretical amount of water displacement of drainage facilities is defined as $Q_{cmax}$, and $Q_{cmax}$ is related to the setting of highway drainage facilities. The drainage capacities are different because of the various timely maintenance of drainage facilities. To elucidate the change degree of facility drainage capacities, the performance evolution coefficient $a$ is introduced, and the value of $a$ is 1 for the new drainage facilities. In general, the value of $a$ is less than or equal to 1, which is relevant to the timeline of drainage facility maintenance. The actual drainage capacity of the facility is labeled as $aQ_{cnax}$.

In this case, the volume of road surface water is the difference in the actual displacement of rainfall drainage facilities:

$$V = Q - 3600 a Q_{cmax} \tag{14}$$

where $V$, $Q$, $a$, and $Q_{cmax}$, represent the volume of the accumulated water on road (m$^3$/h), amount of rainfall (m$^3$/h), performance evolution coefficient of drainage facilities, and theoretical amount of water displacement of drainage facilities (m$^3$/s), respectively.

In addition, different road water accumulation models can be obtained owing to various road alignments and combinations; accordingly different water film depths can be observed. Considering the actual road conditions, the volume of road water is equal to that of different geometric models. Therefore, the model of the water film depth in the case of the road with drainage facilities of poor capacities can be expressed as

$$V_{ji}(H) = Q - 3600 a Q_{cmax} \tag{15}$$

where $V_{ji}(H)$, $Q$, $a$, and $Q_{cmax}$ are the corresponding amount of the accumulated water to different road geometry models (m$^3$/h, number of 1, 2, 3, or 4 for variable i), amount of rainfall

(m$^3$/h), performance evolution coefficient of drainage facilities, and the designed amount of water displacement of drainage facilities (m$^3$/s), respectively.

**2.4.3. Road without drainage capacities.** With the increase in the service life of highway, changes have occurred in the geometric alignment indexes and the road drainage capacity, resulting in a road section with no drainage capacities, which remains a challenge for driving safety during rainfall. In this case, all the rainfall will be collected on the road surface. Rainfall is equal to the amount of the accumulated water, and the corresponding depth of water film model can be expressed as

$$V_{ji}(H) = Q \tag{16}$$

where the $V_{ji}(H)$, $Q$ are the corresponding amount of the accumulated water to different road geometry models (m$^3$/h, number of 1, 2, 3, or 4 for variable i), amount of rainfall (m$^3$/h).

## 3. Results

The verification of the model in this study is divided into two situations. The first is the situation with drainage facilities and normal use, and the difference between the model prediction results is verified by comparison with the existing regression model. By using field experiments, data on rainfall intensity, catchment width, catchment length, and water film thickness are collected, and the prediction accuracy of the theoretical and regression models are analyzed to verify the feasibility and accuracy of the theoretical prediction model. The detailed description and field tests data are presented in S1 File. The second is to select the typical data of the variables in the model when the drainage facilities cannot be used normally or without drainage facilities, as well as conduct a qualitative analysis of the model's rationality.

### 3.1. Verification of the water film depth model for the road with drainage facilities and in normal use

**3.1.1. Verification by comparison with existing regression models.** Presently, the existing water film depth regression prediction models mainly include RRL, Anderson, Gallaway and Ji models, as shown in Table 2.

When the drainage facilities are in normal use, the water film depth prediction formula (13) in this study is compared with the existing regression models, as presented in Fig 7. The value of flow path length is more than 0 and The highway texture depth is generally required to be greater than 0.55mm. Where the length of flow path is 20m and the texture depth is 1 mm. To maintain the consistency of the slope values in the above regression model, the influence of the longitudinal slope on the resultant slope should be reduced. Where the longitudinal slope is 0.5%, and the cross slope is 2%. It can be approximated that the cross slope is equal to the resultant slope.

It can be seen from the **Fig 7** that when the drainage length and slope are constant, the increasing trend of water film depth in present work is similar to that of the RRL model. When the rainfall intensity is 20mm/h, the water film depth calculated by the model in this paper is

**Table 2. Water film depth prediction models.**

| Source | Equation Form | Variables |
|---|---|---|
| RRL [13] | $d = 0.017 \times (L \times I)0.47 \times S^{-0.2}$ | water film depth $d$ length of flow path $L$ rainfall intensity $I$ slope $S$ texture depth $TD$ |
| Anderson [14] | $d = 0.15 \times (L \times I)^{0.5} \times S^{-0.5}$ | |
| Gallaway [15] | $d = 3.38 \times 10^{-3} \times TD^{0.11} \times L^{0.43} \times I^{0.59} \times S^{-0.42} - TD$ | |
| Ji [20] | $d = 0.1258 \times TD^{0.7261} \times L^{0.6715} \times I^{0.7786} \times S^{-0.3147}$ | |

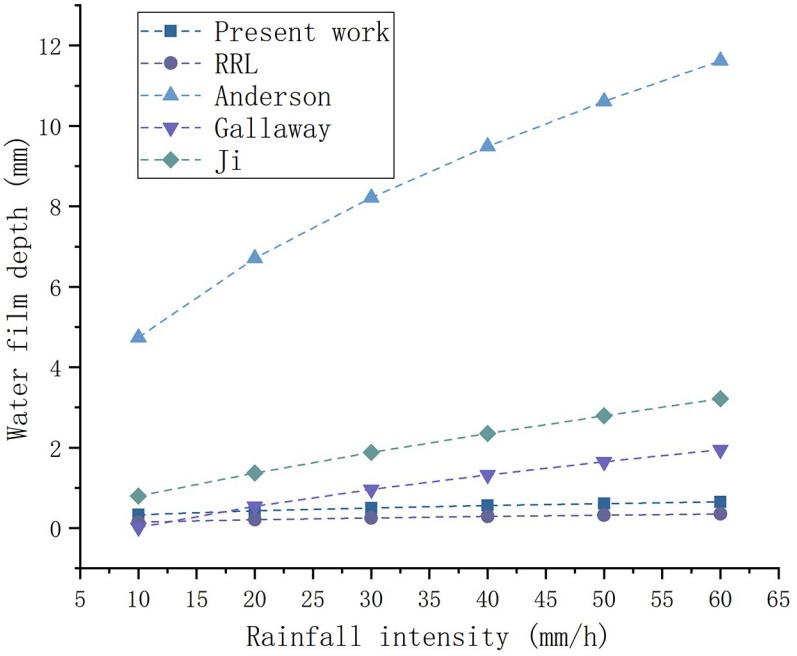

**Fig 7. Comparison of prediction values among different water film depth models.**

close to the value calculated by RRL and Gallaway. When the rainfall intensity is greater than 20mm/h, the calculation results in this study are between the RRL and Gallaway model. The comparison results indicate that the theoretical model can be used to predict the water film depth of the road surface. (2)Verification by field tests.

Between 2019 and 2020, the field measurements of water film depth in rainy days were conducted on several highways in Shaanxi, Shandong, and Zhejiang Provinces to further compare the prediction accuracy of the theoretical model and the regression model, as illustrated in Fig 8.

In rainy days, the measured rainfall intensity, catchment length, and catchment width were 1.2 mm/h, 5 m, and 4 m, respectively. In addition, the cross and longitudinal slopes were both

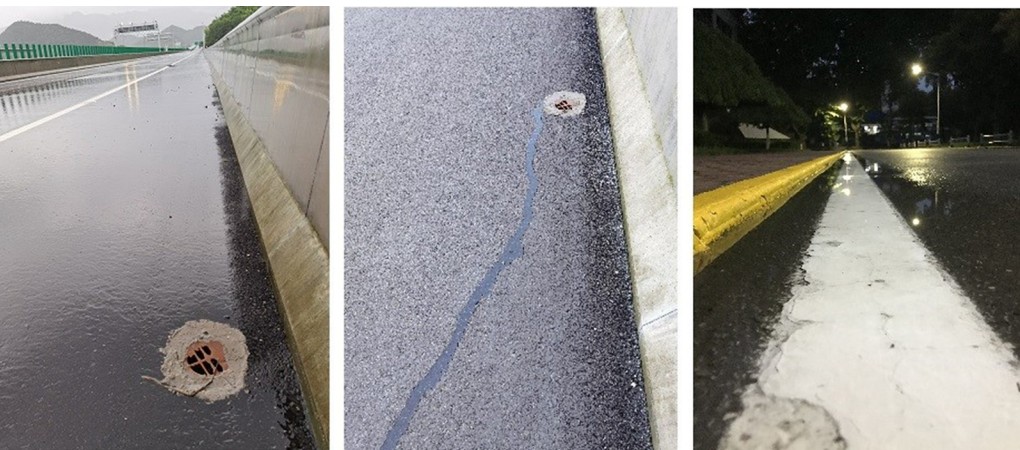

**Fig 8. Field tests.**

**Table 3. Errors between the measured and present work.**

| Measured (mm) | Present work (mm) | Error (%) |
|---|---|---|
| 1.4 | 1.55 | 11 |

2%. The prediction results of the theoretical and Anderson models were compared with the measured data. The error between the measured and present work is presented in Table 3.

From Table 3, it can be observed that compared with the actual measurement data of water film depth, the theoretical model prediction error is 11%. The possible reason for this error is that the water film depth is influenced by external conditions such as road material penetration, evaporation, temperature, and humidity. The theoretical model does not consider the impact of road characteristics (such as material, texture depth).

## 3.2. Model analysis of water film depth when drainage facilities cannot be used normally or without drainage facilities

Owing to untimely maintenance, the drainage facilities may be partially blocked, resulting in the inability to discharge the road surface water in time, which will affect the water film depth prediction model. To revise these results, this study proposes the performance evolution coefficient of drainage facilities. The correction factor needs to be determined based on field tests. Because the experimental support project is a novel construction, its drainage facilities exhibit optimal performance without poor drainage; therefore, targeted experimental verification cannot be conducted. The following qualitatively demonstrates the rationality of the model by altering the values of the variables and analyzing the changes in the prediction results.

According to the design resources and field investigation of Binlai Expressway in the Shandong Province, and the Hangshaotai Expressway in Zhejiang Province, the data in Table 4 are adopted to analyze the models. The principles for determining the data are as follows:

a. The length and width of the catchment should be determined in accordance with the actual resultant gradient of road. For a common four-lane expressway in China, there is a 11.25 m gap between the central isolation belt and the hard shoulder; hence, a catchment width of 11.25 m is adopted here.

b. The distance between drainage facilities is generally set to be 25–50m. When there are many lanes on expressways and first-class highways, a smaller distance should be adopted. In addition, the drainage facilities should be distributed more densely at the bottom of the concave vertical curve. Therefore, the catchment length is selected as 25 m in this study.

c. Considering that a small resultant road gradient facilitates water accumulation better, both cross and vertical slopes are taken as 2%.

**Table 4. Parameter value.**

| Parameter name | Symbol | Unit | Value |
|---|---|---|---|
| Catchment length | $L$ | m | 25 |
| Catchment width | $W$ | m | 11.25 |
| cross slope | $i_h$ | % | 2 |
| Roughness coefficient | $n$ | \ | 0.013 |
| Hydraulic gradient | $I$ | % | 2 |
| Rainfall intensity | $I_R$ | mm/h | 2, 4, 8, 20, 45 |
| Radius of vertical curve | $R$ | m | 15000 |

d. Regarding the common asphalt pavement, the roughness coefficient of the asphalt pavement used in this study is 0.013.

e. Concerning the design data of the Shandong Binlai Expressway and Hangshaotai Expressway, the radius of the vertical curve is 15 000 m here.

f. Rainfall intensity is assigned according to the grade of the short-time rainfall table, as presented in Table 5.

According to the rainfall amount within 12 h or 24 h, the meteorological department usually classifies the rainfall levels into light rain, moderate rain, heavy rain, rainstorm, heavy storm, and disastrous rainstorm. However, in China, there is no uniform classification standard for short-term (1 h) rainfall levels for situations where there is a large amount of rainfall in a short-time, which may exceed the drainage capacity of the drainage facilities, thus accumulating water on the road and affecting driving safety. Based on the field measurement of the rainfall duration of several groups within 60 min, the grade of short-time rainfall is proposed, as presented in Table 5. By comparison with the rainfall intensity levels published by the local meteorological bureau on the same day, it can be determined that the short-term rainfall levels are consistent with the official data, which can be used to provide a theoretical basis to optimally link the rainfall levels with the transportation industry in the future.

1. Model analysis of water film depth when drainage capacity is insufficient
When the drainage facilities with insufficient drainage capacity are installed on a road, the amount of rainfall will be greater than the water displacement of the drainage facilities, thereby triggering an incomplete discharge of rainfall, which results in road water accumulation. Factors such as type, location, and spacing of drainage facilities will influence the designed theoretical drainage capacities of facilities. The drainage capacities of facilities are different owing to the different timeliness of maintenance; hence, it is necessary to investigate the performance evolution coefficient of drainage facilities in combination with field tests to provide support for the water film depth calculation.

2. Analysis of simulation results obtained from water film depth model when the pavement has no drainage capacity

When there is no drainage capacity, all the rainfall will be collected on the road surface, and the rainfall is equal to the amount of accumulated water. Owing to the different road alignments and combinations, different water film depths with a 1 h rainfall duration on a road surface are obtained. To study the influence of the cross slope (1%, 2%, 3%, and 4%) on the water film depth, other parameters of the model should be unchanged. The values of the parameters are listed in Table 4. Fig 9 presents the depth of water film after the rainfall lasts for 1 h for the road section with optimal cross drainage and poor longitudinal drainage (Model A). When the

**Table 5. Grade of short-time rainfall.**

| grade | rainfall intensity (mm) | | | |
|---|---|---|---|---|
| | 5 min | 10 min | 30 min | 60 min |
| Short-time light rain | <0.2 | <0.3 | <0.7 | <2.0 |
| Short-time moderate rain | 0.2–0.5 | 0.3–0.8 | 0.7–1.8 | 2.0–4.0 |
| Short-time heavy rain | 0.5–1.2 | 0.8–2.0 | 1.8–4.0 | 4.0–8.0 |
| Short-time rainstorm | 1.2–4.0 | 2.0–6.0 | 4.0–12.0 | 8.0–20.0 |
| Short-time heavy storm | 4.0–10.0 | 6.0–15.0 | 12.0–30.0 | 20.0–45.0 |

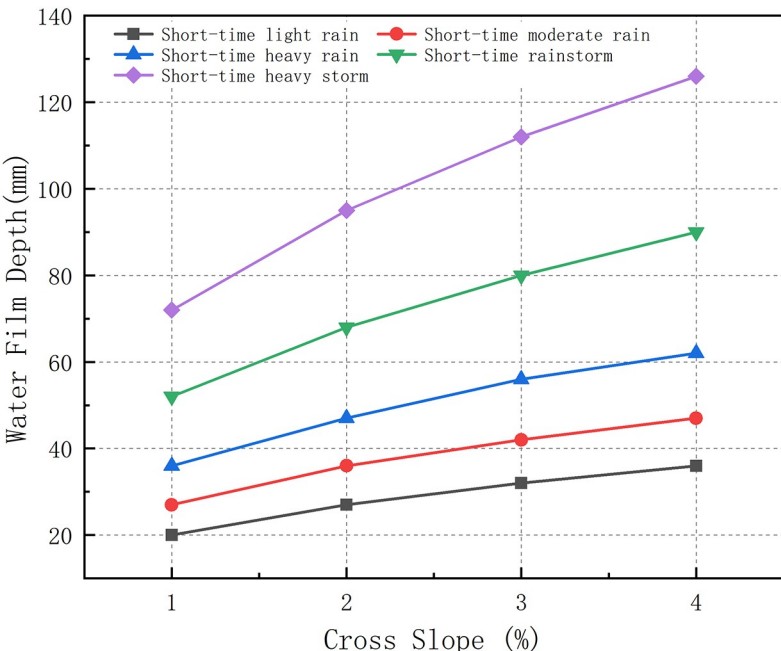

**Fig 9. Cross slope dependence of the depth of water film (Model A).**

parameters of the vertical curve radius, catchment length, and width remain unchanged, the water film depth increases with the increase in the cross slope at the same rainfall level, and the water film depth increases as the rainfall grade increases under the same cross slope.

For Models B, C, and D, Fig 10 illustrates the influence of rainfall intensity (2 mm/h, 4 mm/h, 8 mm/h, 20 mm/h, and 45 mm/h) on the water film depth. It can be inferred that the water

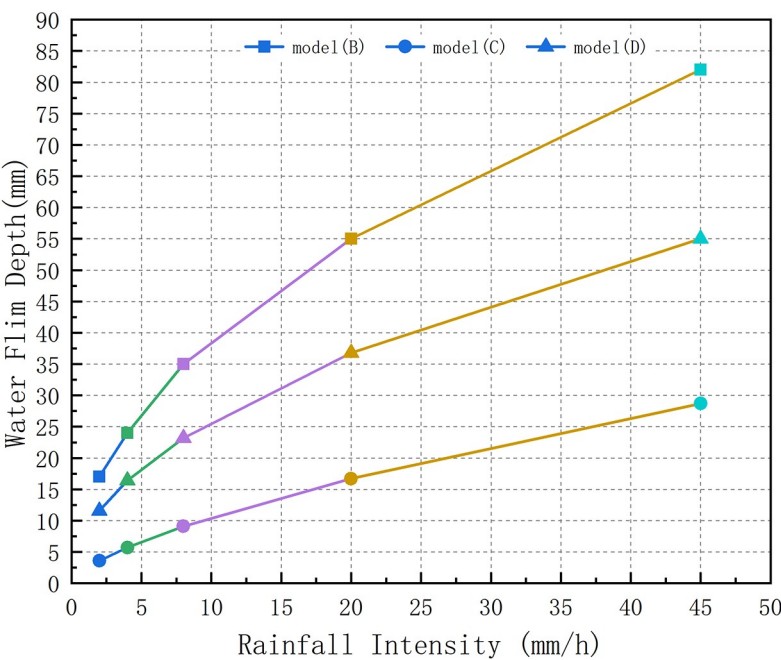

**Fig 10. Relationship between the water film depth and rainfall intensity.**

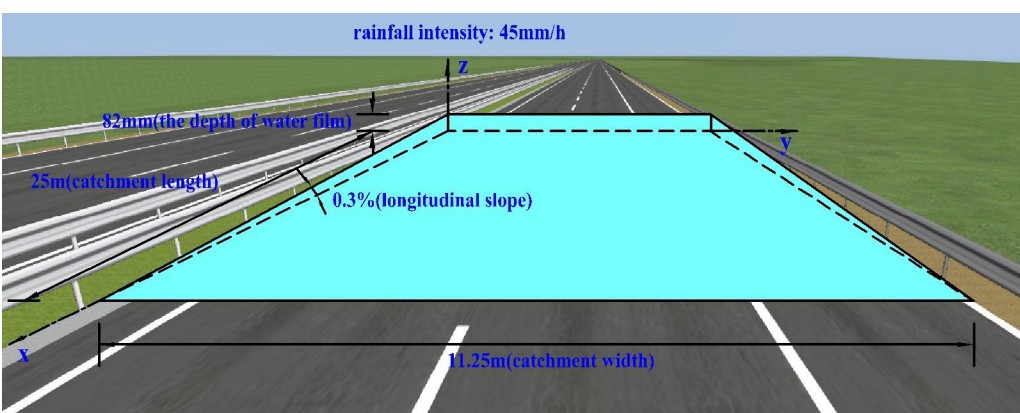

**Fig 11. Schematic of the water film depth (Model B).**

film depth increases with the increase in rainfall intensity, which corresponds with the conclusions of the RRL, Anderson, and Gallaway models.

To illustrate the water film thickness of Models B, C, and D more vividly, the 3D virtual reality tool conventionally used on highways, namely UC-Win/Road software, was adopted to develop three road models with different characteristics.

When the longitudinal slope is 0.3% and the rainfall intensity is 45 mm/h, the water film depth corresponding to Model B is 82 mm, as shown in Fig 11.

For the road section with poor drainage on both cross and vertical slopes (Model C), when the rainfall intensity is 45 mm/h, the corresponding water film depth is 28.7 mm, as presented in Fig 12.

For the road section with poor drainage in the cross slope and good drainage in the longitudinal slope (Model D), when the cross slope is 0.3% and the rainfall intensity is 45 mm/h, the corresponding water film depth is 55 mm, as shown in Fig 13.

## 4. Discussion and conclusion

The effects of road geometric characteristics and drainage capacity of drainage facilities on water film depth were considered, and a theoretical model for predicting water film thickness under different rainfall intensities was established. Based on the field measurement data, the

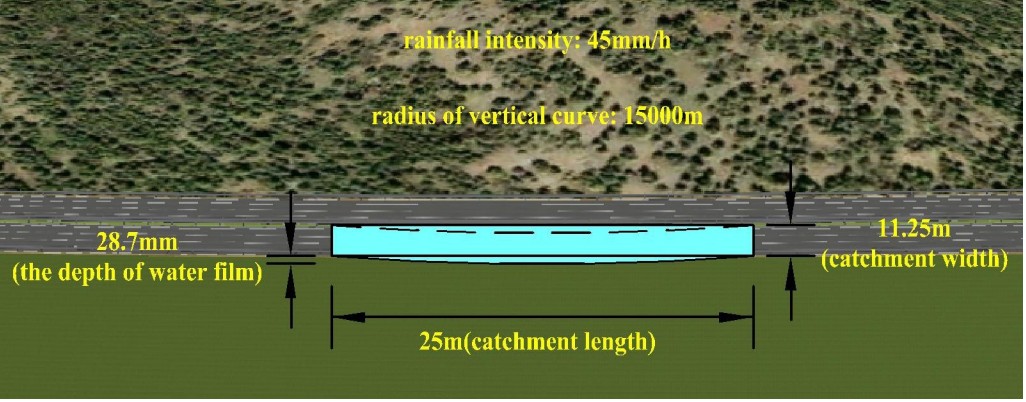

**Fig 12. Schematic of the water film depth (Model C).**

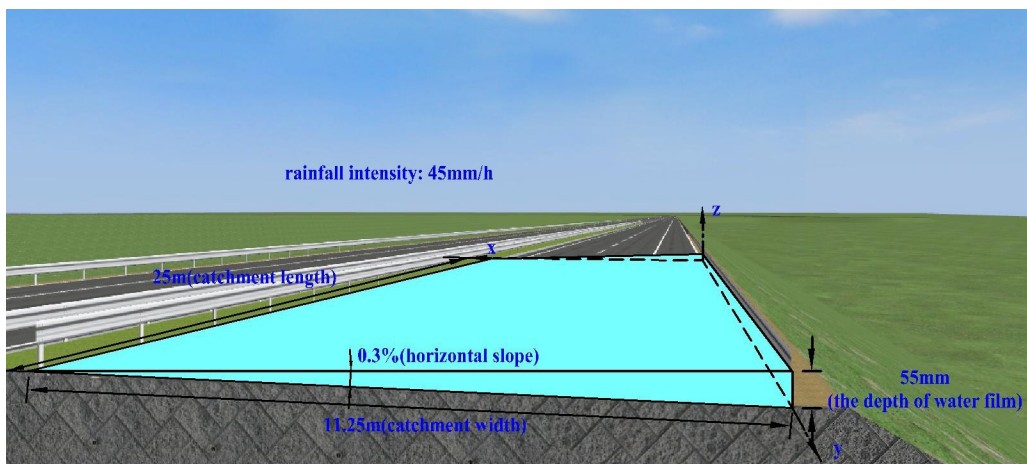

**Fig 13. Schematic of the water film depth (Model D).**

water film thickness prediction model when the road has drainage facilities and in normal use is verified. Compared with the regression model based on a specific experimental data, the theoretical prediction model based on hydraulic theory and mathematical models demonstrated wider applicability. The factors affecting the water film depth in the regression model (slope, rainfall intensity, and drainage length) are considered, as well as the actual road alignment and drainage capacity of drainage facilities.

According to the geometric characteristics of road, the models of road water accumulation are divided into four categories: the road section with optimal cross drainage and poor longitudinal drainage for slope change points of the concave vertical curve included in the section with poor longitudinal drainage (Model A), road section with optimal cross drainage and poor longitudinal drainage for slope change points of the vertical curve not involved in the section with poor longitudinal drainage (Model B), road section with poor drainage in both cross and longitudinal slopes (Model C), and road section with poor drainage in cross slope and good drainage in longitudinal slope (Model D).

Based on the drainage capacities of facilities, the model for the water film depth is divided into three types: the road surface with drainage facilities and in normal use, road with drainage facilities of poor capacities, and road without drainage capacities.

The results indicate that the water film depth exhibits an increasing tendency with the increase in rainfall intensity when parameters such as road alignment and pavement material are constant. For the road section with optimal drainage in the cross slope and poor drainage in longitudinal slope (Model A), the water film depth increases with the increase in cross slop at the same rainfall level when the radius of vertical curve and the catchment length and width remain unchanged, and the water film depth increases as the rainfall grade increases under the same cross slop. Note that the result calculated with the model is slightly larger than that measured in the field on rainy days with an error of 11%, which is probably because of the influence of external conditions, such as penetration, evaporation, temperature, and humidity on the water film depth.

The water film depth of the pavement under different geometric features and drainage capacities of facilities are provided via theoretical modeling. Unlike regression models that solely consider specific parameters, theoretical models can be applied to actual roads where the horizontal and vertical alignment combinations, as well as the capacity of drainage facilities, are ever-changing.

However, there are still some problems yet to be addressed. First, only the flow section with the single cross slope triangle was considered during the calculation for the water displacement of drainage facilities. The prediction model for the water film depth still needs to be verified for drainage facilities with poor capacities. In addition, considering that the timeliness of road maintenance contributes to the variation in the drainage capacity of facilities, field tests are required to investigate the performance evolution of the drainage capacity in future studies. It is necessary to further analyze the sensitive factors that affect the thickness of the water film, which will help to propose solutions to reduce the impact of the water film thickness from the road design stage. The theoretical model ignores the influence of pavement characteristics, and the pavement material correction coefficient can be included in future studies. Moreover, the increase in the water film depth caused by rainfall will reduce the pavement friction coefficient, and further affect the lateral stability of the vehicle and the stopping sight distance. A practical speed limit plan is required to ensure vehicle safety.

## Supporting information

**S1 File. Detailed description and field tests data.**
(PDF)

## Acknowledgments

We are grateful to the Shandong Hi-speed Infrastructure Construction Co., Ltd. for generously providing the highway site and equipment.

## Author Contributions

**Conceptualization:** Shuo Han, Jinliang Xu.

**Data curation:** Shuo Han.

**Investigation:** Shuo Han, Sunjian Gao, Xufeng Li, Zhaoxin Liu.

**Methodology:** Shuo Han, Menghua Yan.

**Resources:** Jinliang Xu.

**Software:** Xunjiang Huang.

**Supervision:** Jinliang Xu.

**Validation:** Jinliang Xu.

**Writing – original draft:** Shuo Han.

**Writing – review & editing:** Shuo Han, Jinliang Xu.

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
