## [Decision Letter · Decision Letter 0]

17 Mar 2021

PONE-D-21-06199

Predicting the Depth of Water Film: A Model Based on the Geometric Features of Road and Drainage Capacities of Facilities

PLOS ONE

Dear Dr. Han,

Thank you for submitting your manuscript to PLOS ONE. After careful consideration, we feel that it has merit but does not fully meet PLOS ONE’s publication criteria as it currently stands. Therefore, we invite you to submit a revised version of the manuscript that addresses the points raised during the review process.

Please consider all the comments.

We look forward to receiving your revised manuscript.

Kind regards,

Ahmed Mancy Mosa, Ph.D.

Academic Editor

PLOS ONE

Journal Requirements:

PLOS requires an ORCID iD for the corresponding author in Editorial Manager on papers submitted after December 6th, 2016. Please ensure that you have an ORCID iD and that it is validated in Editorial Manager. To do this, go to ‘Update my Information’ (in the upper left-hand corner of the main menu), and click on the Fetch/Validate link next to the ORCID field. This will take you to the ORCID site and allow you to create a new iD or authenticate a pre-existing iD in Editorial Manager. Please see the following video for instructions on linking an ORCID iD to your Editorial Manager account: https://www.youtube.com/watch?v=_xcclfuvtxQ

We note that you have indicated that data from this study are available upon request. PLOS only allows data to be available upon request if there are legal or ethical restrictions on sharing data publicly. For more information on unacceptable data access restrictions, please see http://journals.plos.org/plosone/s/data-availability#loc-unacceptable-data-access-restrictions.In your revised cover letter, please address the following prompts:a) If there are ethical or legal restrictions on sharing a de-identified data set, please explain them in detail (e.g., data contain potentially sensitive information, data are owned by a third-party organization, etc.) and who has imposed them (e.g., an ethics committee). Please also provide contact information for a data access committee, ethics committee, or other institutional body to which data requests may be sent.b) If there are no restrictions, please upload the minimal anonymized data set necessary to replicate your study findings as either Supporting Information files or to a stable, public repository and provide us with the relevant URLs, DOIs, or accession numbers. For a list of acceptable repositories, please see http://journals.plos.org/plosone/s/data-availability#loc-recommended-repositories.We will update your Data Availability statement on your behalf to reflect the information you provide.

Reviewers' comments:

Reviewer's Responses to Questions

**Comments to the Author**

1. Is the manuscript technically sound, and do the data support the conclusions?

Reviewer #1: Partly

Reviewer #2: Yes

Reviewer #3: Yes

2. Has the statistical analysis been performed appropriately and rigorously? 

Reviewer #1: N/A

Reviewer #2: I Don't Know

Reviewer #3: I Don't Know

3. Have the authors made all data underlying the findings in their manuscript fully available?

Reviewer #1: Yes

Reviewer #2: Yes

Reviewer #3: Yes

4. Is the manuscript presented in an intelligible fashion and written in standard English?

Reviewer #1: No

Reviewer #2: Yes

Reviewer #3: Yes

5. Review Comments to the Author

Reviewer #1: 1.In general, your English writing needs to improve. Make sure that readers can understand your paper without difficulties. Your abstract has so many grammar mistakes.

2.Literature Review (in your Introduction section) also needs to improve, please read more papers and learn how to write it, please figure out outstanding opinions.

3.Please clarify the innovations, and explain the difference between your work and others, high light your contributions. You can add these in the section of Introduction.

4. In section Rainfall, you give the equation, if this quote from others, please give the reference. Otherwise, please explain how you get this equation. The other equations also need explanations.

5.You have given several assumptions in this paper, please explain why this is reasonable. Also the explanations of the four models you choose in this paper.

6.You may give a case study to show your model, and some GIS figures will be more visible for showing the depth of water on roads.

Reviewer #2: 1、Abstract: Setting a need for research in the depth of water film shall have background data. If data on accidents happening due to poor road conditions is presented then that will help in understanding the gravity of situation and need of the study.

2、If there is a relationship between the depth of water film and the driving risk level, it is better to put forward the driving risk level according to the prediction results of the model, so as to put forward the safety control measures in rainy days.

3、I think F in formula ([Disp-formula pone.0252767.e001]) is redundant.

4、Line 208, the format of “(JTGT_D33-2012)” is incorrect.

Reviewer #3: 1、Lin 215. “Calculation of the Depth of Water Film Depth” should be “Calculation of the Water Film Depth”.

2、Whether the formula or model involved in the paper is based on the others research results? are they all derived by yourself?

3、The model in this paper does not take pavement materials (e.g. bituminous or cement) into account, which can be discussed in future research work, such as adding correction parameters to improve the model.

4、As you mentioned, the grade of short time rainfall is proposed, but where it is, I have not found, the specific standard should be given.

6. PLOS authors have the option to publish the peer review history of their article (what does this mean?). If published, this will include your full peer review and any attached files.

Reviewer #1: No

Reviewer #2: No

Reviewer #3: No

---

## [Author Response · Author response to Decision Letter 0]

13 Apr 2021

TO Reviewer #1:

We appreciate your time and help in reviewing our manuscript. Thank you very much for your affirmation of our work in this paper and many thanks for your detailed comments. We have revised the paper very carefully according to your suggestion, and all the modified contents have been marked in red font in the updated manuscript.

Response:

1.In general, your English writing needs to improve. Make sure that readers can understand your paper without difficulties. Your abstract has so many grammar mistakes.

Response:

Many thanks for your suggestion. 

The English of the full manuscript has been made improvement in the following aspects:

1.We have improved the language expression logic to guide the readers better.

2.We have reorganized language expressions for the verbose passages to make it easier for readers to understand.

3.We have revised the grammar and typos in the original text.

4.We have enriched vocabulary expressions.

5.We have tried to avoid the use of first-person pronouns.

We reorganized and compiled the abstract as follows. 

The depth of water film is a key variable to affect traffic safety under rainfall. Figure out the correlation among the depth of water film and the geometric features of road and drainage capacities of facilities is essential for road safety under a rainy day. First, the road geometry feature was classified into four types, and the facility drainage capacities were concerned from three aspects in this work. Further, the concept of short-time rainfall grade was proposed according to the results of the field test. Finally, the prediction model for the depth of water film based on the geometric features of road and drainage capacity of facilities with different rainfall intensities was conceived. Results show that the depth of water film increases with the increase in rainfall intensity in the case of keeping some parameters constant such as road alignment and materials. Under such circumstance that the drainage facilities on highway are in normal service, the error of the model for depth of water film is 0.15mm when compared with the data measured from the field test on rainy days. Compared with the existing model results, the difference is not significant. This prediction model can be used to evaluate the depth of water film for highway on rainy days. This prediction model can be used to evaluate the depth of water film for highway on rainy days, offering guidance on the measurement of road friction coefficient and the arrangement of security controls on rainy days.

2.Literature Review (in your Introduction section) also needs to improve, please read more papers and learn how to write it, please figure out outstanding opinions.

Response:

Many thanks for pointing out this issue.

We reorganized and compiled the Literature Review as follows.

Researches have shown that the wet and slippery road accompanied with the decrease of friction caused by rainfall is one of the key variables in traffic accidents [1-4]. According to the statistics from the Korean Transportation Safety Authority, although the total number of deaths due to the road traffic accidents decreased in 2013-2016 in Korea, the number of deaths related to rainfall increased from 430 in 2013 to 463 in 2016. Lee et al. concluded that rain and water depth factors and road factors were mutually correlated with the level of accident severity [1]. Research by Brodsky et al. showed that the accident rate on wet and slippery roads on rainy days was 2 to 3 times higher than that on dry roads on sunny days. Moreover, the longer stopping sight distance needed due to a reduction in friction between the tiers and the wet road surface might be more challenging on curves. In a 1980 study conducted by the National Transportation Safety Board, it was found that in the United States, the risk of fatal accidents on wet roads is 3.9 to 4.5 times that on dry roads [3]. Research by Qiu et al. showed that rainfall led to an average increase of 71% in traffic accident rates, which had a significant impact on traffic accidents [4]. Yu et al. found that the probability of a collision increased significantly for drivers traveling through downgrade steep segments under large precipitation conditions. The study also proved that the rain collision rate is related to different road sections and road geometric characteristics [5].To find out the causes of high incidence of road traffic accidents on rainy days, researchers focused on the relationship between the friction coefficient and the depth of water film, and reached the consensus that there was a direct correlation between the reduction of friction coefficient and the depth of water film on road surface [6-8]. Through indoor water spray experiments, Do et al. measured the friction coefficient under water depth (<1mm), and the friction force-water depth curve was conformed to the Stribeck curve [6]. A finite element simulation model was established to predict the friction coefficient under different water film depth, and found that the friction coefficient decreased with the increase of water film depth at the same driving speed [7].Using a dynamic friction coefficient tester, dynamic friction coefficient test experiments were carried out under different water film depth through artificial rainfall experiments and found that the decrease in friction coefficient is directly related to the depth of water film on the pavement [8]. The thicker the water film is, the worse the skid resistance is. Even worse, the hydroplaning phenomenon may occur, imposing severe impact on the driving safety under rainfall [9-11]. Hence, a prediction model for depth of water film is vital in determining the road friction coefficient and managing the security controls on rainy days. As is known, the highway traffic environment is complex in the rain, various geometric features of road and drainage capacities of facilities have different influences on the depth of water film, and so does the different rainfall intensities. Luo et al. used variance analysis and multi-factor treatment statistical methods, and concluded that the calculation of water film depth is affected by rainfall (rainfall intensity, duration) and pavement characteristics (cross slope, longitudinal slope, pavement material properties, texture, and penetration), and the flow path slope has a greater impact on surface drainage than the length of the flow path [12].Concerning such complex factors as aforementioned, numerous investigations on the depth of water film have been carried out, and different prediction models have been developed. However, most prediction models were established based on the in-door simulation experiments, which is ill-considered for the actual road circumstances and external environmental conditions such as wind, causing it being limited in practical applications. For example, based on the in-door rainfall simulation experiment, the UK Road Research Laboratory proposed the typical RRL model, i.e., the depth of rain water relating drainage length, rainfall intensity (5 minutes duration), and slope [13]. B. M. Gallaway et al developed the model of the effects of rainfall intensity, pavement cross slope, surface texture, and drainage length on water depth [14]. On the basis of Gallaway model, R. S. Huebner et al built up the PAVDRN computer model, which presented that rainfall intensity (5 minutes duration), pavement section geometry and structure accounted for the depth of water film [15]. Based on the classical RRL and Gallaway models, the water film depth regression model was calibrated according to the data from test by J. Luo et al [16-18]. With the Gallaway model framework, J. Chesterton et al gave a forecast of the depth of water film for New Zealand highway [19]. And using the collision data in Florida, W. Jayasooriya et al. evaluated the prediction precision of the Gallaway and PAVDRN models for the depth of water film and hydroplaning speed [20].

Clearly, the models mentioned above are in accordance with some specific test data in their experience. However, because of the three-dimensional structures of the actual roads, the combinations of road slope, superelevation and alignment are complex and variable to adapt to the terrain variation and engineering requirements. The drainage capacities of the drainage facilities will undergo some changes in their service lives. Taking such factors into consideration, the effects of such complex factors on the depth of pavement water film may not be illustrated clearly using the models available now. Therefore, the parameters cannot be modified according to the actual situation, imposing restrict on the fields of applications for them.

3.Please clarify the innovations, and explain the difference between your work and others, high light your contributions. You can add these in the section of Introduction.

Response:

Many thanks for pointing out this issue.

To explain the difference between our work and others, highlight our contributions, we added two main advantages of this study in the section of Introduction as follows. 

To overcome the limitations of the aforementioned methods, and thus give an accurate prediction for the correlation between the depth of water film and the geometric features of road and drainage capacities of facilities, we proposed a prediction model for water film depth in theory. More specifically, the road geometry feature was considered from four types and the facility drainage capacities were concerned from three aspects in this work, and the rainfall intensity involved short time light rain, moderate rain, heavy rain, rainstorm, and heavy storm. And the prediction model for the depth of water film based on the geometric features of road and drainage capacity of facilities with different rainfall intensities was developed. The two main advantages of this paper are as follows: First, the study puts forward a classification of short-term rainfall. Second, the model established at the theoretical level clarifies the influence of road geometric characteristics and drainage capacity of drainage facilities on the depth of water film. The results can provide guidance on the measurement of road friction coefficient and the arrangement of security controls on rainy days.

4. In section Rainfall, you give the equation, if this quote from others, please give the reference. Otherwise, please explain how you get this equation. The other equations also need explanations.

Response:

Many thanks for pointing out this problem.

Formula 11 is derived from drainage design specifications (Ministry of communications of china, Design Specification for Highway Alignment, PRC Industry Standard JTG D20-2017). The reference has been added to the text, and the rest of the formulas are derived from the perspective of mathematical models. The formulas are explained in detail below.

The geometric models for water film depth with different road geometric characteristics are established based on the different combinations of cross slope and longitudinal slope, and the different convergence directions of slop flow. The amount of gathered water is the volume of the gathered water in the geometric model, and the water film depth is the depth in the geometric model. From the perspective of mathematical models, formulas are used in this paper to derive the volume and water film depth with different geometric models.

During rainfall, this work focuses on the situation of setting water retaining belts or edge stones, such as Shandong Binlai Expressway. Because of the different combinations of road alignment and the effect of road alignment, water retaining belt or edge stone on rainfall, the rainfall will be collected on the highway surface, forming slope flow and then being discharged from the drainage facilities. The amount of rainfall is equal to the volume of converged slope flow in unit time.

5.You have given several assumptions in this paper, please explain why this is reasonable. Also the explanations of the four models you choose in this paper.

Response:

Many thanks for your question. We elaborated this issue as follows:

When the water is gathered on the surface, there will be water film with a certain of depth, which is the basic idea of the water film depth prediction model. The amount of surface gathered water is defined as the amount of rainfall subtracts the amount of drainage, which means that the volume of surface gathered water in unit time is the difference between the volume of rainfall and drainage. Despite the same volume of gathered water, the different road geometry features and convergence directions of flow cause different depths of water film. The amount of rainfall is related to the rainfall intensity, and the amount of drainage is bound up with the facility drainage capacities. Therefore, the prediction model for water film depth is proposed here based on the geometric features of road and drainage capacity of facilities with different rainfall intensities.

For the purpose of field verification and model derivation, the pavement water film depth in rainy days mentioned here is defined as the depth of water film under short-time (1h) rainfall. Compared with the highway without water retaining edge stone such as Hangshaotai Expressway, the rainfall will converged on the road surface, forming slop flow and then being discharged from the drainage facilities for the highway with water retaining belt or edge stone. Therefore, this work focuses on the depth of water film at the outside edge of hard shoulder in the case of setting water retaining belts or edge stones. The road surface water film depth is related to the amount of accumulated water, which refers to the difference between the amount of rainfall and the water displacement of drainage facilities. Albeit with the same water accumulation, various road alignments and combinations result in different depths of water film. Hence, the construction of the model about the depth of water film includes the calculation of rainfall, construction of road water accumulation model, calculation of water displacement of drainage facilities, and derivation of formulas about road surface water film depth. 

A prediction model for water film depth is proposed theoretically in this paper. And to meet the requirements of topographic changes and engineering, the effects of rainfall evaporation, infiltration and runoff, and the pavement material properties are ignored.

The reason for water gathering on surface locates in the road section with poor drainage. Despite the same volume of gathered water, the different road geometry features and convergence directions of flow cause different depths of water film. According to the values of horizontal and longitudinal gradients, there are 4 types for the road section with poor drainage, following with 4 geometric models for road water accumulation.

The geometric models for water film depth with different road geometric characteristics are established based on the different combinations of cross slope and longitudinal slope, and the different convergence directions of slop flow. The amount of gathered water is the volume of the gathered water in the geometric model, and the water film depth is the depth in the geometric model. 

6.You may give a case study to show your model, and some GIS figures will be more visible for showing the depth of water on roads.

Response:

Thank you very much for your suggestion and expanding our knowledge.

Due to time issues, we chose the familiar UC-Win/Road simulation software, which is a 3D virtual reality tool widely used on highways. We wonder if it can meet your expectations. The UC-Win/Road simulation software was used to build 3 highway models of different geometric characteristics. In order to better describe the model, we added a few figures generated by UC-Win/Road simulation software and AutoCAD, as shown below.

For the road section with good horizontal drainage and poor longitudinal drainage (Model B), the depth of water film increases with the increase in rainfall intensity when the catchment length, catchment width and longitudinal gradient remain unchanged. Fig 10 shows the depth of water film of road with rainfall lasting for 1h. Given the catchment length of 25m, catchment width of 11.25m, longitudinal gradient of 0.3% and rainfall intensity of 46 mm/h under the short-time heavy storm, the corresponding depth of water film is 83mm, as shown in Figure 12. 

Fig 12. Schematic diagram of the depth of water film (Model B).

For the road section with poor drainage in both horizontal and longitudinal slope (Model C), when keeping the radius of vertical curve, catchment length and catchment width unchanged, the depth of water film will increase as rainfall intensity increases. As shown in Fig 10, the depth of water film is 29.2mm in the case of water catchment length of 25m, catchment width of 11.25m, radius of vertical curve of 15000m and rainfall intensity of 46mm/h under the short-time heavy storm, as shown in Figure 13. 

Fig 13. Schematic diagram of the depth of water film (Model C).

For the road section with poor drainage in cross slope and good drainage in longitudinal slope (Model D), when no changes occur in the catchment length, catchment width and longitudinal gradient, the depth of water film will increase with the increase in rainfall intensity. As shown in Fig 10, under the condition of short time heavy storm, a depth of water film of 55.8mm can be obtained corresponding to the situation with the catchment length of 25m, catchment width of 11.25m, horizontal gradient of 0.3%, and rainfall intensity of 46mm/h, as shown in Figure 14. 

Fig 14. Schematic diagram of the depth of water film (Model D).

To Reviewer #2:

Thank you very much for the detailed comments. We appreciate your time and help in reviewing our manuscript, and the insightful comments you provided that have helped significantly improve the quality of this study. We have revised the paper very carefully according to your suggestions, and detailed explanations of all the issues are as follows.

Response:

1.Abstract: Setting a need for research in the depth of water film shall have background data. If data on accidents happening due to poor road conditions is presented then that will help in understanding the gravity of situation and need of the study.

Response:

Many thanks for pointing out this problem.

To better understand research need, background data has been added in Introduction section.

Researches have shown that the wet and slippery road accompanied with the decrease of friction caused by rainfall is one of the key variables in traffic accidents [1-4]. According to the statistics from the Korean Transportation Safety Authority, although the total number of deaths due to the road traffic accidents decreased in 2013-2016 in Korea, the number of deaths related to rainfall increased from 430 in 2013 to 463 in 2016. Lee et al. concluded that rain and water depth factors and road factors were mutually correlated with the level of accident severity [1]. Research by Brodsky et al. showed that the accident rate on wet and slippery roads on rainy days was 2 to 3 times higher than that on dry roads on sunny days. Moreover, the longer stopping sight distance needed due to a reduction in friction between the tiers and the wet road surface might be more challenging on curves. In a 1980 study conducted by the National Transportation Safety Board, it was found that in the United States, the risk of fatal accidents on wet roads is 3.9 to 4.5 times that on dry roads [3]. Research by Qiu et al. showed that rainfall led to an average increase of 71% in traffic accident rates, which had a significant impact on traffic accidents [4]. Yu et al. found that the probability of a collision increased significantly for drivers traveling through downgrade steep segments under large precipitation conditions. The study also proved that the rain collision rate is related to different road sections and road geometric characteristics [5].

2.If there is a relationship between the depth of water film and the driving risk level, it is better to put forward the driving risk level according to the prediction results of the model, so as to put forward the safety control measures in rainy days.

Response:

Thank you very much for your suggestion and expanding my knowledge. 

Researchers conducted works about the water film depth and driving risk. Typical representatives include the researches of Ahmed, Do M.-T., Peng J. [R1-R3]. The water film depth has impact on road friction coefficient, which affects the lateral driving stability. Too small friction coefficient may cause slippage and overturning of driving, and increase the braking distance when braking, causing safe velocity lower than the standard driving speed and raising the driving risk. However, a constant friction coefficient is adopted for safe velocity calculation in current specifications such as AASHTO, which loses sight of the effect of pavement water film depth. Unfortunately, we do not discuss such a point in this paper because of the workload. And we will make a further study it in next steps. Thanks again for your suggestions.

References cited in the response letter here for your convenience:

[R1] Mohamed M. Ahmed, Ghasemzadeh A . The impacts of heavy rain on speed and headway Behaviors: An investigation using the SHRP2 naturalistic driving study data[J]. Transportation Research Part C: Emerging Technologies, 2018, 91:371-384.

[R2] Do M.-T., Cerezo V., Beautru Y., Kane M. Modeling of the connection road surface microtexture/water depth/friction. Wear, 2013; 302(1-2): 1426-1435.

[R3] Peng J., Chu L., Fwa T. F. Determination of safe vehicle speeds on wet horizontal pavement curves. Road Materials and Pavement Design. 2020. DOI: 10.1080/14680629.2020.1772350.

3.I think F in formula (1) is redundant.

Response:

Many thanks for pointing out this problem. We reinterpreted the F (catchment area) as follows.

The pavement water film depth in rainy days is defined as the depth of water film at the outside edge of hard shoulder under short-time (1h) rainfall. During rainfall, this work focuses on the situation of setting water retaining belts or edge stones, such as Shandong Binlai Expressway. Because of the different combinations of road alignment and the effect of road alignment, water retaining belt or edge stone on rainfall, the rainfall will be collected on the highway surface, forming slope flow and then being discharged from the drainage facilities, distance between which is usually set to be 25~50m. Therefore, the effect of catchment area (F) is involved in the calculation in this work. 

When the water is gathered on the surface, there will be water film with a certain of depth, which is the basic idea of the water film depth prediction model. The amount of surface gathered water is defined as the amount of rainfall subtracts the amount of drainage, which means that the volume of surface gathered water in unit time is the difference between the volume of rainfall and drainage. Despite the same volume of gathered water, the different road geometry features and convergence directions of flow cause different depths of water film. The amount of rainfall is related to the rainfall intensity, and the amount of drainage is bound up with the facility drainage capacities. Therefore, the prediction model for water film depth is proposed here based on the geometric features of road and drainage capacity of facilities with different rainfall intensities.

In this work, the water film depth refers to that on the highway with water retaining belt or edge stone. Typical representative for the case of overflow drainage, i.e., no water retaining edge stones on pavement, is the Hangzhou Shaoxing Taizhou Expressway. In the case of the pavement with good drainage and ignoring the effects of such factors as pavement texture, material properties and converge flow, it can be considered roughly as that the real-time water film depth during rainfall is equal to the rainfall intensity. 

4.Line 208, the format of “(JTGT_D33-2012)” is incorrect.

Response:

Many thanks for pointing out this issue. 

We have revised the format as follows: “(JTG/T D33-2012)”.

To Reviewer #3:

We appreciate your time and help in reviewing our manuscript. Thank you very much for your affirmation of our work in this paper and many thanks for your detailed comments. We have revised the paper very carefully according to your suggestion, and all the modified contents have been marked in red font in the updated manuscript.

Response:

1.Lin 215. “Calculation of the Depth of Water Film Depth” should be “Calculation of the Water Film Depth”.

Response:

Many thanks for pointing out this issue. 

We have revised the heading as follows: “Calculation of the Water Film Depth”.

2.Whether the formula or model involved in the paper is based on the others research results? are they all derived by yourself?

Response:

Formula 11 is derived from drainage design specifications (Ministry of communications of china, Design Specification for Highway Alignment, PRC Industry Standard JTG D20-2017). The reference has been added to the text, and the rest of the formulas and models are derived from the perspective of mathematical models. The formulas and models are explained in detail below.

The geometric models for water film depth with different road geometric characteristics are established based on the different combinations of cross slope and longitudinal slope, and the different convergence directions of slop flow. The amount of gathered water is the volume of the gathered water in the geometric model, and the water film depth is the depth in the geometric model. From the perspective of mathematical models, formulas are used in this paper to derive the volume and water film depth with different geometric models.

During rainfall, this work focuses on the situation of setting water retaining belts or edge stones, such as Shandong Binlai Expressway. Because of the different combinations of road alignment and the effect of road alignment, water retaining belt or edge stone on rainfall, the rainfall will be collected on the highway surface, forming slope flow and then being discharged from the drainage facilities. The amount of rainfall is equal to the volume of converged slope flow in unit time.

3.The model in this paper does not take pavement materials (e.g. bituminous or cement) into account, which can be discussed in future research work, such as adding correction parameters to improve the model.

Response:

Thank you very much for your suggestion and expanding our knowledge. 

Due to the workload, the influence of pavement materials is not considered in the model. In this work, the amount of rainfall is defined as the volume of rainwater accumulated on the water surface without evaporation, infiltration and runoff, which comes from the rainwater falling from the sky to the ground in the case of a certain catchment area. However, it could not be ignored that the pavement texture and material properties have a great influence on the water film depth. This is also the work to be studied in the future.

4.As you mentioned, the grade of short time rainfall is proposed, but where it is, I have not found, the specific standard should be given.

Response:

Many thanks for your suggestion. The grade of short time rainfall is presented in Table 3 in Results.

Based on the field measurement of the rainfall duration of several groups within 60min, the grade of short time rainfall is proposed, as shown in Table 5. By comparing with the rainfall intensity levels published by the local meteorological bureau on the same day, it can be found that the short-term rainfall levels are consistent with the official data, which can be used to provide a theoretical basis for better linking the rainfall levels with the transportation industry in the future. 

Table5. The grade of short time rainfall

grade rainfall intensity (mm)

 5 min 10 min 30 min 60 min

Short time light rain ＜0.2 ＜0.3 ＜0.7 ＜2.0

Short time moderate rain 0.2~0.5 0.3~0.8 0.7~1.8 2.0~4.0

Short time heavy rain 0.5~1.2 0.8~2.0 1.8~4.0 4.0~8.0

Short time rainstorm 1.2~4.0 2.0~6.0 4.0~12.0 8.0~20.0

Short time heavy storm 4.0~10.0 6.0~15.0 12.0~30.0 20.0~45.0

---

## [Decision Letter · Decision Letter 1]

21 Apr 2021

PONE-D-21-06199R1

Predicting the Depth of Water Film: A Model Based on the Geometric Features of Road and Drainage Capacities of Facilities

PLOS ONE

Dear Dr. Han,

Thank you for submitting your manuscript to PLOS ONE. After careful consideration, we feel that it has merit but does not fully meet PLOS ONE’s publication criteria as it currently stands. Therefore, we invite you to submit a revised version of the manuscript that addresses the points raised during the review process.

Please, carefully consider the comments of the reviewer. Be accurate please.

We look forward to receiving your revised manuscript.

Kind regards,

Ahmed Mancy Mosa, Ph.D.

Academic Editor

PLOS ONE

Reviewers' comments:

Reviewer's Responses to Questions

**Comments to the Author**

1. If the authors have adequately addressed your comments raised in a previous round of review and you feel that this manuscript is now acceptable for publication, you may indicate that here to bypass the “Comments to the Author” section, enter your conflict of interest statement in the “Confidential to Editor” section, and submit your "Accept" recommendation.

Reviewer #1: (No Response)

Reviewer #2: (No Response)

Reviewer #3: (No Response)

2. Is the manuscript technically sound, and do the data support the conclusions?

Reviewer #1: Partly

Reviewer #2: Yes

Reviewer #3: Yes

3. Has the statistical analysis been performed appropriately and rigorously? 

Reviewer #1: No

Reviewer #2: Yes

Reviewer #3: Yes

4. Have the authors made all data underlying the findings in their manuscript fully available?

Reviewer #1: Yes

Reviewer #2: Yes

Reviewer #3: Yes

5. Is the manuscript presented in an intelligible fashion and written in standard English?

Reviewer #1: (No Response)

Reviewer #2: Yes

Reviewer #3: Yes

6. Review Comments to the Author

Reviewer #1: 1. Your English writing needs to improve. The Abstract and Literature Review have been improved, while they need further improvement, as well as other sections. Don't repeat same sentence:“This prediction model can be used to evaluate the depth of water film for highway on rainy days. This prediction model can be used to evaluate the depth of water film for highway on rainy days, offering guidance on the measurement of road friction coefficient and the arrangement of security controls on rainy days.”

2. Literature Review :Similar statements made by different authors have not been integrated, resulting in repetition of facts and a lack of cohesion and logical flow. Please rewrite sentences like “Moreover, the longer stopping sight distance needed due to a reduction in friction between the tiers and the wet road surface might be more challenging on curves.”.

3. “we proposed a prediction model for water film depth in theory.” However, when I entered “prediction model for water film depth” in science direct, there are so much papers, please clarify the corrections and difference between your work and previous studies.

4. Your models come from others (Ministry of communications

of china, Design Specification for Highway Alignment, PRC Industry Standard JTG D20-2017). So please clarify your work and others, and why you choose the modes, which part is others and which is your contributions.

5. Please give explanations in Table 2. Parameter value. How did you get these data, from experiments or reference papers? Same problems in your work with other data.

6.This paper is to predict the depth of water film, your results are based on what data? How to evaluate the reliability of predict results？The results compared with models that you proposed, which is a bias.

Reviewer #2: For the first comment, you better add a typical background data in abstract. For the second comment, if the safety control measures is not put forword, it seems make this work less significant, or the author at least mentions it in future work.

Reviewer #3: The author has addressed the comments well. But for the third comment, the author gave a full explanation, but I didn't find corresponding change in the conclusion of the paper.

7. PLOS authors have the option to publish the peer review history of their article (what does this mean?). If published, this will include your full peer review and any attached files.

Reviewer #1: No

Reviewer #2: No

Reviewer #3: No

---

## [Author Response · Author response to Decision Letter 1]

7 May 2021

TO Reviewer #1:

We appreciate your time and help in reviewing our manuscript. Thank you very much for your affirmation of our work in this paper and many thanks for your detailed comments. We have revised the paper very carefully according to your suggestion, and all the modified contents have been marked in red font in the updated manuscript.

Response:

1. Your English writing needs to improve. The Abstract and Literature Review have been improved, while they need further improvement , as well as other sections. Don't repeat same sentence:“This prediction model can be used to evaluate the depth of water film for highway on rainy days. This prediction model can be used to evaluate the depth of water film for highway on rainy days, offering guidance on the measurement of road friction coefficient and the arrangement of security controls on rainy days.”

Response:

Many thanks for your suggestion. To improve the level of English writing, we have checked the full text of repetitive sentences and changed the expression of repetitive sentences. In addition, professional editing agencies have been invited to help improve English writing.

2. Literature Review :Similar statements made by different authors have not been integrated, resulting in repetition of facts and a lack of cohesion and logical flow. Please rewrite sentences like “Moreover, the longer stopping sight distance needed due to a reduction in friction between the tiers and the wet road surface might be more challenging on curves.”

Response:

Many thanks for pointing out this issue.We reorganized and compiled the Literature Review as follows. 

Studies have demonstrated that the wet and slippery nature of roads, accompanied with the decrease in friction, caused by rainfall is a key variable in traffic accidents [1–5]. Evidently, with less friction, the braking distance of the vehicle increases at the same speed, which triggers frequent collisions in rainy weather. In their research, Brodsky et al. demonstrated that the accident rate on wet and slippery roads on rainy days was 2–3 times higher than that on dry roads on sunny days. Moreover, the longer stopping sight distance required due to a reduction in friction between tiers and the wet road surface might be more challenging on curves [3]. To further study the reasons for the reoccurring rain collision accidents caused by the reduction in the friction coefficient, researchers focused on the relationship between the friction coefficient and the water film depth, and reached the consensus that there was a direct correlation between the reduction in friction coefficient and the water film depth on road surface [6–8]. Studies inferred that there are primarily two methods of regression analysis and simulation adopted in studying the relationship between the friction coefficient and the water film depth. Using artificial rainfall, Luo et al. carried out dynamic friction coefficient test experiments under different water film depth and determined that the reduction in the friction coefficient is directly related to the water film depth of the road surface [6]. In addition, via indoor water spray experiments, Do et al. determined that the relationship between the friction coefficient and the water film depth (less than 1 mm) agrees with the Stribeck curve [7]. Unlike regression analysis, the finite element simulation model was established to predict the friction coefficient under different water film depths, and it was deduced that at the same driving speed, the friction coefficient decreased with the increase in the water film depth [8]. Although the research methods differ, they all arrived at a consistent conclusion: The thicker the water film, the lower the friction coefficient. Further, the hydroplaning phenomenon may occur, which adversely impacts driving safety under rainfall conditions [9–11].

Hence, a prediction model for the water film depth is crucial in determining the road friction coefficient and evaluating the impact of rainfall on traffic safety. However, various geometric features of road and facility drainage capacities, as well as different rainfall intensities, exert different influences on the water film depth. Luo et al. adopted the variance analysis and multi-factor treatment statistical methods and concluded that the calculation of water film depth is affected by rainfall (rainfall intensity and duration) and pavement characteristics (cross slope, longitudinal slope, pavement material properties, texture, and penetration) [12]. Regarding the aforementioned complex factors, several investigations on the water film depth have been conducted, such as regression analysis and artificial neural network [13–16]. Among them, the most widely adopted method is the water film depth regression model established via experiments. For example, based on the in-door rainfall simulation experiment, the UK Road Research Laboratory proposed the conventional RRL model, i.e., the rain water depth relative to drainage length, rainfall intensity (5-min duration), and slope [13]. Anderson modified the parameters of the RRL model and proposed a water film depth prediction model for a plane impermeable surface [14]. Unlike the RRL model, which only considers two pavement texture depths (1.8 mm and 2.4 mm), Gallaway developed a water film depth prediction model for the U.S. Department of Transportation, which considers nine pavement texture depths. The relationship between rainfall intensity, cross slope, surface texture, drainage-path length, and water film depth are determined through artificial rainfall experiments [15]. The RRL, Anderson, and Gallaway models are simple and widely used by other researchers for developing water film depth prediction models [17–18], regression model verification [19–20], water film depth prediction [21–23], etc.

3.“we proposed a prediction model for water film depth in theory.” However, when I entered “prediction model for water film depth” in science direct, there are so much papers, please clarify the corrections and difference between your work and previous studies.

Response:

Many thanks for pointing out this issue.

We have emphasized the corrections and difference with other models in the Introduction and Conclusion respectively.

(1)Introduction:

Compared with the RRL, Anderson, and Gallaway models, the theoretical model established in this study considers the geometric characteristics of the real road environment and the drainage capacity of drainage facilities; therefore, it can accurately reveal the main factors affecting the water film depth and describe the influence of the rainfall intensity, road geometric characteristics, and facility drainage capacities on the water film thickness. Hence, the proposed model overcomes the limitations of regression models.

(2)Conclusion:

Compared with the regression model based on a specific experimental data, the theoretical prediction model based on hydraulic theory and mathematical models demonstrated wider applicability. The factors affecting the water film depth in the regression model (slope, rainfall intensity, and drainage length) are considered, as well as the actual road alignment and drainage capacity of drainage facilities.

4. Your models come from others (Ministry of communications of china, Design Specification for Highway Alignment, PRC Industry Standard JTG D20-2017). So please clarify your work and others, and why you choose the modes, which part is others and which is your contributions.

Response:

Many thanks for pointing out this problem.We have carefully studied and made further modifications in the process of model establishment and derivation. The issue is explained as follows.

(1) Interpretation of normative references:

The criterion (Ministry of communications of china, Design Specification for Highway Alignment, PRC Industry Standard JTG D20-2017) only clarifies the vertical slope that is prone to water accumulation on the road surface and the value range of the composite slope. This paper establishes four road water accumulation models based on the range of slope values and the combination of cross and longitudinal slopes. 

(2) Formula source:

This article uses a formula in the specification (Formula 11), which is the calculation formula for the shallow triangular discharge capacity formula of a single cross slope, and reference is marked in the text. The rest includes the use of AutoCAD to draw a schematic diagram of the road area water model, and establishing the calculation formula of water film depth.

Now the revised are listed partly as follows:

Page 7:

Even though the same volume of water is gathered, the different road geometry features and convergence directions of flow ensure different water film depths. According to the gradient values of cross and longitudinal slopes, the sections with poor drainage are divided into four types. Correspondingly, from the geometric perspective, four geometric models for road surface water are established by the AutoCAD software, as presented in Table 1. Road surface water accumulation and water film depth refer to the volume and height of water, respectively, in the geometric model.

Page 8:

The geometric models of water film depths with different road geometric characteristics are established based on different combinations of cross and longitudinal slopes. To obtain the volume and height of the geometric model, i.e., road surface water accumulation and water film depth, it is necessary to further derive the volume and water depth calculation formulas of different geometric models. Subsequently, the derivation process of the mathematical formula is described in detail.

Page 13:

The calculation of the displacement adopts the shallow triangular discharge capacity formula of a single cross slope specified by the Specifications for Drainage Design of Highway (JTG/T D33-2012) [25], and it is expressed as.

Page 14:

The real-time water film depth of the road surface during rainfall is the water depth of the cross section. Based on Equation 12, the water film depth model can be derived as.

Page 15:

In this case, the volume of road surface water is the difference in the actual displacement of rainfall drainage facilities.

5.Please give explanations in Table 2. Parameter value. How did you get these data, from experiments or reference papers? Same problems in your work with other data.

Response:

Many thanks for your question. 

According to different data sources, we reorganized the results section of the paper. 

The verification of the model in this study was categorized into two situations. One was the situation with drainage facilities and normal operation, where the model prediction results were verified by comparing them with results from the existing regression model. Through field experiments, data pertaining to rainfall intensity, catchment width, catchment length, and water film thickness were obtained, and the prediction accuracies of the theoretical and regression models were analyzed to verify the feasibility and accuracy of the theoretical prediction model. The detailed description and field test data are shown in S1 Text in the Supporting Information.

In the second situation, typical data of the variables in the model when the drainage facilities cannot be operated normally or do not have drainage facilities were selected, and the rationality of the model was analyzed qualitatively. Based on the design resources and field investigation of the Binlai Expressway in Shandong Province and the Hangshaotai Expressway in Zhejiang Province, the principles for determining the data used to analyze the models are elaborated.

The principles for determining the data are as follows:

(a) The length and width of the catchment should be determined in accordance with the actual resultant gradient of road. For a common four-lane expressway in China, there is a 11.25 m gap between the central isolation belt and the hard shoulder; hence, a catchment width of 11.25 m is adopted here.

(b) The distance between drainage facilities is generally set to be 25–50m. When there are many lanes on expressways and first-class highways, a smaller distance should be adopted. In addition, the drainage facilities should be distributed more densely at the bottom of the concave vertical curve. Therefore, the catchment length is selected as 25 m in this study.

(c) Considering that a small resultant road gradient facilitates water accumulation better, both cross and vertical slopes are taken as 2%.

(d) Regarding the common asphalt pavement, the roughness coefficient of the asphalt pavement used in this study is 0.013.

(e) Concerning the design data of the Shandong Binlai Expressway and Hangshaotai Expressway, the radius of the vertical curve is 15 000 m here.

(f) Rainfall intensity is assigned according to the grade of the short-time rainfall table, as presented in Table 6.

6.This paper is to predict the depth of water film, your results are based on what data? How to evaluate the reliability of predict results？The results compared with models that you proposed, which is a bias.

Response:

Thank you very much for your question.The model validation in the article has been reorganized and modified, and the three questions are explained below respectively.

(1)Data

The data source is divided into two parts: measured data and selected data. The measured data can be viewed through Supporting information. The principle of data selection has been added and elaborated in the text.

(2)The reliability of predict results

The verification of the model in this study was categorized into two situations. 

One was the situation with drainage facilities and normal operation, where the model prediction results were verified by comparing them with results from the existing regression model. Through field experiments, data pertaining to rainfall intensity, catchment width, catchment length, and water film thickness were obtained, and the prediction accuracy of the theoretical and regression models was analyzed to verify the feasibility and accuracy of the theoretical prediction model.

In the second situation, typical data of the variables in the model when the drainage facilities cannot be operated normally or do not have drainage facilities were selected, and the rationality of the model was analyzed qualitatively.

(3)The results compared with models that our proposed

Extensive citation of the model has already been stated in the Introduction. 

The RRL, Anderson, and Gallaway models are simple and widely used by other researchers for developing water film depth prediction models [17–18], regression model verification [19–20], water film depth prediction [21–23], etc.

The basis for the model selection are explained before verification.

Currently, the most widely used water film depth regression prediction models are the RRL, Anderson, and Gallaway models. The RRL and Gallaway models consider the state of road water flow as sheet flow, without considering the effects of drainage facilities. However, this study considers roads with drainage facilities, which are typical on highways in China. In this study, unlike the flooding state, the water flow is first converged under the effects of the water-retaining belt and curb, and then discharged from the outlet. The water film depth is the water depth in the water-retaining zone or curb. The Anderson model considers impervious pavements, and the theoretical model used in this study disregards the penetration of the pavement; therefore, the Anderson model was used for comparative analysis.

To Reviewer #2:

Thank you very much for the detailed comments. We appreciate your time and help in reviewing our manuscript, and the insightful comments you provided that have helped significantly improve the quality of this study. We have revised the paper very carefully according to your suggestions, and detailed explanations of all the issues are as follows.

Response:

For the first comment, you better add a typical background data in abstract. For the second comment, if the safety control measures is not put forword, it seems make this work less significant, or the author at least mentions it in future work

Response:

Many thanks for pointing out this problem. Modifications have been made based on your suggestions.

(1)Typical background data

Water film depth is a key variable that affects traffic safety under rainfall. According to the Federal Highway Administration (FHWA), approximately 5700 people are killed and more than 544 700 people are injured in crashes on wet pavement annually. 

References

FHWA, 2021. Road Weather Management Program [Access date: 25/4/2021]. https://ops.fhwa.dot.gov/Weather/weather_events/rain_flooding.htm

(2)The safety control measures have been added it in future work.

It is necessary to further analyze the sensitive factors that affect the thickness of the water film, which will help to propose solutions to reduce the impact of the water film thickness from the road design stage. The theoretical model ignores the influence of pavement characteristics, and the pavement material correction coefficient can be included in future studies. Moreover, the increase in the water film depth caused by rainfall will reduce the pavement friction coefficient, and further affect the lateral stability of the vehicle and the stopping sight distance. A practical speed limit plan is required to ensure vehicle safety.

To Reviewer #3:

We appreciate your time and help in reviewing our manuscript. Thank you very much for your affirmation of our work in this paper and many thanks for your detailed comments. We have revised the paper very carefully according to your suggestion, and all the modified contents have been marked in red font in the updated manuscript.

Response:

The author has addressed the comments well. But for the third comment, the author gave a full explanation, but I didn't find corresponding change in the conclusion of the paper.

Response:

Many thanks for pointing out this issue. Modifications have been made based on your suggestions .

However, there are still some problems yet to be addressed. First, only the flow section with the single cross slope triangle was considered during the calculation for the water displacement of drainage facilities. The prediction model for the water film depth still needs to be verified for drainage facilities with poor capacities. In addition, considering that the timeliness of road maintenance contributes to the variation in the drainage capacity of facilities, field tests are required to investigate the performance evolution of the drainage capacity in future studies. It is necessary to further analyze the sensitive factors that affect the thickness of the water film, which will help to propose solutions to reduce the impact of the water film thickness from the road design stage. The theoretical model ignores the influence of pavement characteristics, and the pavement material correction coefficient can be included in future studies. Moreover, the increase in the water film depth caused by rainfall will reduce the pavement friction coefficient, and further affect the lateral stability of the vehicle and the stopping sight distance. A practical speed limit plan is required to ensure vehicle safety.

---

## [Decision Letter · Decision Letter 2]

11 May 2021

PONE-D-21-06199R2

Predicting the Water Film Depth: A Model Based on the Geometric Features of Road and Capacity of Drainage Facilities

PLOS ONE

Dear Dr. Han,

Thank you for submitting your manuscript to PLOS ONE. After careful consideration, we feel that it has merit but does not fully meet PLOS ONE’s publication criteria as it currently stands. Therefore, we invite you to submit a revised version of the manuscript that addresses the points raised during the review process.

Please consider the comments of Reviewer 1.

We look forward to receiving your revised manuscript.

Kind regards,

Ahmed Mancy Mosa, Ph.D.

Academic Editor

PLOS ONE

Journal Requirements:

Reviewers' comments:

Reviewer's Responses to Questions

**Comments to the Author**

1. If the authors have adequately addressed your comments raised in a previous round of review and you feel that this manuscript is now acceptable for publication, you may indicate that here to bypass the “Comments to the Author” section, enter your conflict of interest statement in the “Confidential to Editor” section, and submit your "Accept" recommendation.

Reviewer #1: (No Response)

Reviewer #2: All comments have been addressed

Reviewer #3: All comments have been addressed

2. Is the manuscript technically sound, and do the data support the conclusions?

Reviewer #1: Partly

Reviewer #2: Yes

Reviewer #3: Yes

3. Has the statistical analysis been performed appropriately and rigorously? 

Reviewer #1: Yes

Reviewer #2: Yes

Reviewer #3: Yes

4. Have the authors made all data underlying the findings in their manuscript fully available?

Reviewer #1: Yes

Reviewer #2: Yes

Reviewer #3: Yes

5. Is the manuscript presented in an intelligible fashion and written in standard English?

Reviewer #1: No

Reviewer #2: Yes

Reviewer #3: Yes

6. Review Comments to the Author

Reviewer #1: 1.Your English writing is improved. While the Abstract needs further improvement.

2.The corrections and differences between your work and previous studies are not enough.

3.The results should be compared with more previous studies.

Reviewer #2: The author has tried to address all the comments well, and the modifications are adequate. I recommend acceptance of this paper.

Reviewer #3: The author has addressed all the comments by the futher modification. I consider to accept this paper.

7. PLOS authors have the option to publish the peer review history of their article (what does this mean?). If published, this will include your full peer review and any attached files.

Reviewer #1: No

Reviewer #2: No

Reviewer #3: No

---

## [Author Response · Author response to Decision Letter 2]

19 May 2021

TO Reviewer #1:

We appreciate your time and help in reviewing our manuscript. Thank you very much for your affirmation of our work in this paper and many thanks for your detailed comments. We have revised the paper very carefully according to your suggestion, and all the modified contents have been marked in red font in the updated manuscript.

Response:

1.Your English writing is improved. While the Abstract needs further improvement.

Response:

Many thanks for your suggestion. We reorganized and compiled the Abstract as follows. In addition, professional editing agencies have been invited to help improve English writing.

The water film depth is a key variable that affects traffic safety under rainfall conditions. According to the Federal Highway Administration, approximately 5700 people are killed and more than 544 700 people are injured in crashes on wet pavements annually. While several studies have attempted to address water film depth issues by establishing prediction models, a few focused on the relationship among road geometric features, capacity of drainage facilities and water film depth. To ascertain the influence of the geometric features of road and facility drainage capacities on the water film depth, the road geometry features were first classified into four types, and the facility drainage capacities were considered from three aspects in this study. Furthermore, the concept of short-time rainfall grade was proposed according to the results of the field test. Finally, the theoretical prediction model for the water film depth was conceived, based on the geometric features of road and facility drainage capacities with different rainfall intensities. Compared with the traditional regression prediction models, the theoretical prediction model clearly shows the effects of the geometric features of road and facility drainage capacities. When the road drainage facilities have no drainage capacity, the water film depth increases rapidly with the rainfall intensity. This model can be used to predict the water film depth of road surfaces on rainy days, evaluate the effect of rainfall on the driving environment, and provide guidance for determining safety control measures on rainy days.

2. The corrections and differences between your work and previous studies are not enough.

Response:

Many thanks for pointing out this issue.We reorganized and emphasized the corrections and difference with previous studies in Abstract, Introduction and Discussion and Conclusion respectively as follows. 

(1)Abstract

While several studies have attempted to address water film depth issues by establishing prediction models, a few focused on the relationship among road geometric features, capacity of drainage facilities and water film depth.

(2)Introduction

Clearly, the aforementioned regression models are in accordance with some specific test data in their experience. Their applications are limited to the types of parameters covered by the models, and when the parameter values are within the scope of the database used in their derivation. Owing to the changeable combination of road slope, superelevation, and alignment, facility drainage capacities and environment, the regression model may not accurately describe the influence of various factors on the pavement water film depth.

To address the limitations of the above approaches, this study intends to establish a prediction model for the water film depth, with a full theoretical foundation and extensive application scope to clearly demonstrate the effect of the main parameters. The theoretical model for predicting the water film depth under different rainfall intensities is established by considering four geometry features of roads and three kinds of facility drainage capacities to predict the relationship among the water film depth, the geometric characteristics of the road and the facility drainage capacities accurately.

Compared with the regression models, the theoretical model established in this study considers the geometric characteristics of the real road environment and the drainage capacity of drainage facilities; therefore, it can accurately reveal the main factors affecting the water film depth and describe the influence of the rainfall intensity, road geometric characteristics, and facility drainage capacities on the water film thickness. Hence, the proposed model on a more theotical basis overcomes the limitations of regression models.

(3)Discussion and Conclusion

Compared with the regression model based on a specific experimental data, the theoretical prediction model based on hydraulic theory and mathematical models demonstrated wider applicability. The factors affecting the water film depth in the regression model (slope, rainfall intensity, and drainage length) are considered, as well as the actual road alignment and drainage capacity of drainage facilities.

3.The results should be compared with more previous studies.

Response:

Many thanks for pointing out this issue. We have added more previous studies to compare with the predicted results of this study.

Presently, the existing water film depth regression prediction models mainly include RRL, Anderson, Gallaway and Ji models, as shown in Table 1.

Table 1. Water Film Depth Prediction Models

Source Equation Form Variables

RRL water film depth d

length of flow path L

rainfall intensity I

slope S

texture depth TD

Anderson 

Gallaway 

Ji 

When the drainage facilities are in normal use, the water film depth prediction formula (13) in this study is compared with the existing regression models, as presented in Fig 1. The value of flow path length is more than 0 and The highway texture depth is generally required to be greater than 0.55mm. Where the length of flow path is 20m and the texture depth is 1 mm. To maintain the consistency of the slope values in the above regression model, the influence of the longitudinal slope on the resultant slope should be reduced. Where the longitudinal slope is 0.5%, and the cross slope is 2%. It can be approximated that the cross slope is equal to the resultant slope.

Fig 1. Comparison of prediction values among different water film depth models

It can be seen from the Fig 1 that when the drainage length and slope are constant, the increasing trend of water film depth in present work is similar to that of the RRL model. When the rainfall intensity is 20mm/h, the water film depth calculated by the model in this paper is close to the value calculated by RRL and Gallaway. When the rainfall intensity is greater than 20mm/h, the calculation results in this study are between the RRL and Gallaway model. The comparison results indicate that the theoretical model can be used to predict the water film depth of the road surface.

---

## [Decision Letter · Decision Letter 3]

24 May 2021

Predicting the Water Film Depth: A Model Based on the Geometric Features of Road and Capacity of Drainage Facilities

PONE-D-21-06199R3

Dear Dr. Han,

We’re pleased to inform you that your manuscript has been judged scientifically suitable for publication and will be formally accepted for publication once it meets all outstanding technical requirements.

Kind regards,

Ahmed Mancy Mosa, Ph.D.

Academic Editor

PLOS ONE

Additional Editor Comments (optional):

Reviewers' comments:

Reviewer's Responses to Questions

**Comments to the Author**

1. If the authors have adequately addressed your comments raised in a previous round of review and you feel that this manuscript is now acceptable for publication, you may indicate that here to bypass the “Comments to the Author” section, enter your conflict of interest statement in the “Confidential to Editor” section, and submit your "Accept" recommendation.

Reviewer #1: All comments have been addressed

Reviewer #2: All comments have been addressed

Reviewer #3: All comments have been addressed

2. Is the manuscript technically sound, and do the data support the conclusions?

Reviewer #1: Yes

Reviewer #2: Yes

Reviewer #3: Yes

3. Has the statistical analysis been performed appropriately and rigorously? 

Reviewer #1: Yes

Reviewer #2: Yes

Reviewer #3: Yes

4. Have the authors made all data underlying the findings in their manuscript fully available?

Reviewer #1: Yes

Reviewer #2: Yes

Reviewer #3: Yes

5. Is the manuscript presented in an intelligible fashion and written in standard English?

Reviewer #1: Yes

Reviewer #2: Yes

Reviewer #3: Yes

6. Review Comments to the Author

Reviewer #1: (No Response)

Reviewer #2: The author handled my comments well, and I think that this paper may be considered for acceptance.

Reviewer #3: According to the author's modification of the paper, I suggest acceptance of this paper under meeting the requirements of journal publication.

7. PLOS authors have the option to publish the peer review history of their article (what does this mean?). If published, this will include your full peer review and any attached files.

Reviewer #1: No

Reviewer #2: No

Reviewer #3: No

---

## [Editor Report · Acceptance letter]

24 Jun 2021

PONE-D-21-06199R3 

Predicting the Water Film Depth: A Model Based on the Geometric Features of Road and Capacity of Drainage Facilities 

Dear Dr. Han:

I'm pleased to inform you that your manuscript has been deemed suitable for publication in PLOS ONE. Congratulations! Your manuscript is now with our production department. 

Kind regards, 

on behalf of

Dr. Ahmed Mancy Mosa 

Academic Editor

PLOS ONE